# Structure and function of the ROR2 cysteine-rich domain in vertebrate noncanonical WNT5A signaling

**Samuel C Griffiths[1†‡], Jia Tan[2†], Armin Wagner[3], Levi L Blazer[4], Jarrett J Adams[4], Srisathya Srinivasan[2], Shayan Moghisaei[2], Sachdev S Sidhu[4], Christian Siebold[1*], Hsin-Yi Henry Ho[2*]**

[1]Division of Structural Biology, Wellcome Centre for Human Genetics, University of Oxford, Oxford, United Kingdom; [2]Department of Cell Biology and Human Anatomy, University of California, Davis School of Medicine, Davis, United States; [3]Science Division, Diamond Light Source, Harwell Science and Innovation Campus, Didcot, United Kingdom; [4]School of Pharmacy, University of Waterloo, Waterloo, Canada

**Abstract** The receptor tyrosine kinase ROR2 mediates noncanonical WNT5A signaling to orchestrate tissue morphogenetic processes, and dysfunction of the pathway causes Robinow syndrome, brachydactyly B, and metastatic diseases. The domain(s) and mechanisms required for ROR2 function, however, remain unclear. We solved the crystal structure of the extracellular cysteine-rich (CRD) and Kringle (Kr) domains of ROR2 and found that, unlike other CRDs, the ROR2 CRD lacks the signature hydrophobic pocket that binds lipids/lipid-modified proteins, such as WNTs, suggesting a novel mechanism of ligand reception. Functionally, we showed that the ROR2 CRD, but not other domains, is required and minimally sufficient to promote WNT5A signaling, and Robinow mutations in the CRD and the adjacent Kr impair ROR2 secretion and function. Moreover, using function-activating and -perturbing antibodies against the Frizzled (FZ) family of WNT receptors, we demonstrate the involvement of FZ in WNT5A-ROR signaling. Thus, ROR2 acts via its CRD to potentiate the function of a receptor super-complex that includes FZ to transduce WNT5A signals.

## Editor's evaluation

This manuscript describes the crystal structure of the extracellular portion of the ROR2 cell surface receptor, which plays important roles in development and disease. The work provides valuable new insights into the mechanism by which WNTs interact with cell surface receptors to activate downstream signaling events. The insights are clear, and the supporting data are convincing.

## Introduction

ROR proteins make up an important branch of the receptor tyrosine kinase (RTK) superfamily, conserved from sponges to humans. Originally identified as orphan receptors based on sequence homology to other RTKs (hence the name Receptor tyrosine kinase-like Orphan Receptor), work over the past two decades has elucidated a critical role of the ROR RTK family in mediating noncanonical WNT5A signaling (*Oishi et al., 2003*; *Mikels and Nusse, 2006*; *Ho et al., 2012*; *Green et al., 2008*). Unlike canonical WNTs, which signal through β-catenin-dependent transcription to regulate cell proliferation and tissue fate, WNT5A signals noncanonically through β-catenin-independent mechanisms to induce cytoskeletal rearrangements and tissue morphogenetic changes (*Konopelski et al., 2023*; *Moon et al., 1993*; *Veeman et al., 2003*). The pathway is also of clinical significance, as mutations in

**\*For correspondence:**
christian.siebold@strubi.ox.ac.
uk (CS);
hyhho@ucdavis.edu (HHH)

[†]These authors contributed equally to this work

**Present address:** [‡]Evotec (UK) Ltd, Abingdon, United Kingdom

**Competing interest:** The authors declare that no competing interests exist.

WNT5A, the ROR family member ROR2, and the downstream signal transducers Dishevelled 1 (DVL1) and DVL3 have been reported to cause Robinow syndrome (RS), a congenital disorder characterized by systemic tissue shortening defects, including dwarfism, mesomelic limb shortening, brachydactyly, genitourinary defects, cleft palate, and other craniofacial dysmorphisms (*Person et al., 2010*; *Afzal et al., 2000*; *van Bokhoven et al., 2000*; *Bunn et al., 2015*; *White et al., 2015*; *White et al., 2016*), and a distinct cohort of ROR2 missense mutations cause brachydactyly type B (BDB) (*Oldridge et al., 2000*; *Schwabe et al., 2000*). Moreover, elevated expression of ROR1 or ROR2 correlates with increased cancer metastatic potentials, and several anti-ROR therapies are currently in various stages of development (*Rebagay et al., 2012*; *Kipps, 2022*). The etiological mechanisms of these mutations, however, remain largely uncharacterized. Thus, a greater understanding of ROR receptor function is important from both basic science and medical perspectives.

ROR receptors are type-I transmembrane (TM) proteins with a single-pass TM helix linking extracellular and intracellular regions. The extracellular region (ECD) of vertebrate ROR proteins consists of an immunoglobulin (Ig) domain, a Frizzled (FZ)-like cysteine-rich domain (CRD), and a Kringle (Kr) domain. The intracellular region includes a tyrosine kinase domain and a serine/threonine/proline-rich domain (*Minami et al., 2010*; *Green et al., 2008*). The specific requirement of these domains in WNT5A signaling remains controversial. Early genetic studies in *Caenorhabditis elegans* showed that only the CRD and the TM helix are essential for the function of the nematode ROR homolog Cam-1 in cell migration, which raised the possibility that Cam-1 may not act as a typical RTK and may instead regulate the spatial distribution of WNT ligands (*Kim and Forrester, 2003*). Experiments in vertebrate systems, however, largely suggest that ROR proteins act as bona fide WNT signaling receptors and that this function requires other domains of ROR proteins, including the intracellular domains (*DeChiara et al., 2000*; *Oishi et al., 2003*; *Mikels and Nusse, 2006*). However, due to the historical lack of tractable assays to directly measure ROR activity, the precise requirement of vertebrate ROR proteins in noncanonical WNT5A signaling has not been systematically examined.

The CRD is of broader interest because it is not only conserved within the ROR family but also among other important receptor classes where the domain mediates ligand and/or co-factor binding through a signature hydrophobic groove or pocket (*Bazan and de Sauvage, 2009*). For instance, the CRD of the classical WNT receptor FZ interacts with the palmitoleate moiety of WNT ligands directly through this groove (*Janda et al., 2012*). Free fatty acids have also been observed to interact in the same fashion (*Nile et al., 2017*). Moreover, the CRD of the Hedgehog signal transducer and GPCR Smoothened (Smo) binds cholesterol through an analogous hydrophobic pocket (*Byrne et al., 2016*). Because the CRD of ROR2 was previously implicated in WNT5A binding (*Oishi et al., 2003*; *Mikels and Nusse, 2006*), and shares a high degree of amino acid sequence similarity with the FZ CRD (*Xu and Nusse, 1998*; *Saldanha et al., 1998*), it is assumed that it possesses a similar hydrophobic groove via which it interacts with WNT5A. However, this hypothesis remains untested, as the requirement of the vertebrate ROR2 CRD in WNT5A signaling and its atomic structure have not been determined.

In this study, we determined the crystal structure of the ROR2 CRD and Kr domains. Remarkably, we found that the two domains share an extended interface and that the ROR2 CRD lacks the characteristic hydrophobic groove/pocket for interacting with lipids. The latter observation suggests that the ROR2 CRD cannot mediate high-affinity interaction with the palmitoleate group of WNT5A. To further probe the requirement of the ROR2 CRD in WNT5A signaling, we developed a functional complementation assay in *Ror1/Ror2* double knockout mouse embryonic fibroblasts (MEFs) and showed that the ROR2 CRD is required and minimally sufficient to mediate WNT5A-ROR signaling. We implicated the FZ family in the pathway by showing that WNT5A directly interacts with the CRD of multiple FZ proteins, and that synthetic Ig that bind the FZ CRDs can potently activate or inhibit WNT5A-ROR signaling depending on their valency. Lastly, we demonstrated that several Robinow patient mutations in the CRD and Kr domains impair ROR2 secretion and function, presumably by disrupting the proper folding of these domains. Collectively, the study provides structural and functional insights into ROR2 function and supports a model in which ROR2 acts through its CRD to promote FZ-dependent WNT5A signaling.

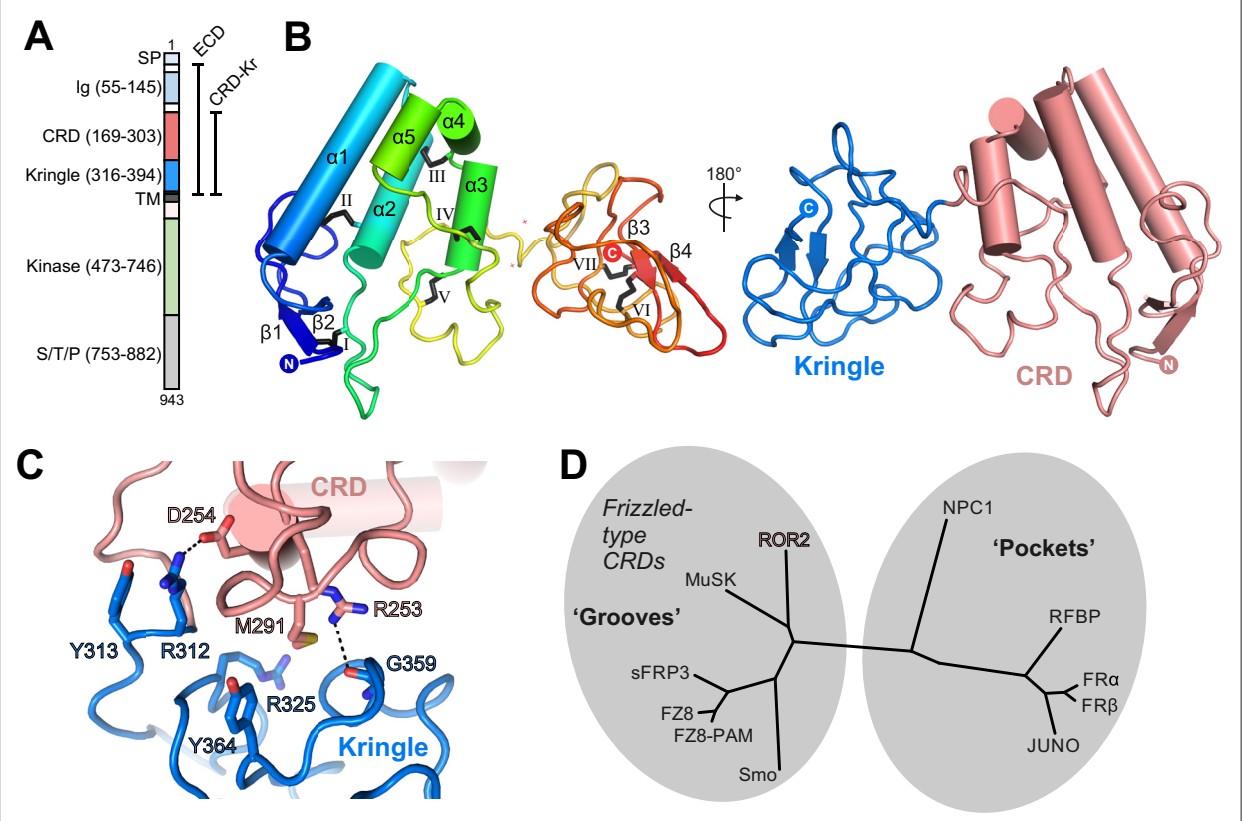

**Figure 1.** Structure of the ROR2 cysteine-rich domain (CRD) and Kringle (Kr) domains. (**A**) Domain layout of ROR2 and constructs used in this study. SP, signal peptide; S/T/P, serine/threonine/proline-rich domain. Other domains are defined in the text. (**B**) Cartoon representation of the ROR2 CRD-Kr structural unit colored in a rainbow representation (N terminus: blue, C terminus: red), with secondary structural elements indicated and disulfide bonds numbered using Roman numerals. The right panel shows a two-domain representation of ROR2, with the CRD in salmon and the Kr domain in blue. (**C**) Close-up view on the ROR2 CRD-Kr interface rotated 90° relative to (**B**). Interface residues are shown in stick representation and color-coded as in B, right panel. Hydrogen bonds are displayed as a dashed line. (**D**) Structural phylogenetic analysis of CRDs, adapted from Figure 5 of *Nachtergaele et al., 2013*, to include ROR2.

The online version of this article includes the following figure supplement(s) for figure 1:

**Figure supplement 1.** ROR2 purification, characterization, and structure solution.

**Figure supplement 2.** Comparison of the ROR2 Kringle (Kr) to other related Kr structures.

## Results

### The structure of the ROR2 CRD and Kr domains

To determine the structure of the ROR2 CRD, we expressed a range of constructs comprising the full-length human ROR2 ECD (*Figure 1A*). Analysis of construct secretion revealed that deletion of the Kr domain severely impacted the yield of ROR2 constructs (*Figure 1—figure supplement 1A*), and therefore the full ECD and CRD-Kr were selected for large-scale expression and purification (*Figure 1—figure supplement 1B and C*).

We determined a crystal structure of the ROR2 CRD-Kr tandem domain construct at a resolution of 2.48 Å via a platinum single-wavelength anomalous dispersion experiment coupled with molecular-replacement (MR-SAD) (*Supplementary file 1*; *Figure 1—figure supplement 1D–H*; see experimental procedures for details). The CRD comprises 5 α-helices (α1–5) and a single β-sheet (strands β1 and β2), while the Kr domain presents a characteristic lack of secondary structure, displaying a single β-sheet (strands β3 and β4) (*Figure 1B*, left-hand panel). The CRD is stabilized by five disulfide bonds: one located between β1 and a loop extending from helix α2 (I), a second linking α2 and the loop preceding longest helix α1 (II), a third between helix α2 and the loop between helices α3/4 (III), a fourth between long loops following helices α2 and α3 (IV), and the fifth between helix α3 itself and the loop extending from α5 (V). The Kr domain is stabilized by two disulfide bonds (VI and VII) that

are found within the core of the Kr domain. A structured linker is observed between both domains, containing one additional free cysteine (C316). This is positioned in proximity to the C-terminus of our expression construct, which contains one additional cysteine residue (C394), unresolved in our ROR2 CRD-Kr crystal structure (*Figure 1—figure supplement 1D*).

Overall, the CRD and Kr domains form an associated structural unit (*Figure 1B*, right-hand panel). A contact interface is observed between the two domains and is dominated by van der Waals interactions, with two hydrogen bonds present (*Figure 1C*). The domains share a small interfacial area of 559 Å², with a reasonable shape complementarity score (0.7) (*Lawrence and Colman, 1993*). The structure of Kr in the CRD-Kr tandem domain closely resembles that previously described for the isolated human ROR1 and ROR2 Krs (*Goydel et al., 2020*; *Guarino et al., 2022*; *Figure 1—figure supplement 2A*). Shi et al. recently described the structure of the ECD of the *Drosophila* ROR2 homolog Nrk, which consists of a CRD and a Kr domain (*Shi et al., 2021*). There is also a high degree of structural conservation between the ROR2 and Nrk Kr domains (*Shi et al., 2021*; *Figure 1—figure supplement 2A*). Kr domains are generally observed to constitute protein-protein interfaces within multi-domain proteins (*Deguchi et al., 1997*; *Ultsch et al., 1998*; *Zebisch et al., 2016*), suggesting that the ROR2 Kr domain acts to stabilize the CRD (*Figure 1—figure supplement 1A*). Interestingly, while the CRD and Kr domains of Nrk also form an associated structural unit (*Shi et al., 2021*), and the secondary structural elements of these domains are highly conserved between human and *Drosophila* (*Figure 1—figure supplement 2B*), the spatial arrangement of CRD and Kr in ROR2 is substantially different from that in Nrk (*Figure 1—figure supplement 2C*). Correspondingly, the CRD-Kr interfaces in ROR2 and Nrk are also structurally distinct (*Figure 1—figure supplement 2C*).

The full-length ROR2 ECD is monomeric in solution at concentrations as high as 48 µM (*Figure 1—figure supplement 1B and C*), indicating that the CRD does not mediate dimerization as has been suggested for other related FZ-type CRDs (*Dann et al., 2001*), and also suggesting that the Ig and Kr domains do not facilitate oligomerization. We conducted small-angle X-ray scattering experiments with in-line size exclusion chromatography (SEC-SAXS) experiments using the full-length ROR2 ECD (*Figure 2—figure supplement 1*). The CRD-Kr arrangement observed in the crystal structure is conserved in solution (*Figure 2—figure supplement 1A*), while the N-terminal Ig domain is flexible relative to the CRD-Kr domain unit (*Figure 2—figure supplement 1B*), as predicted from the 15-residue linker between domains. Additionally, SAXS does not suggest that the ROR2 ECD forms a dimer in solution; however, we cannot exclude the possibility that the TM helix and/or the intracellular

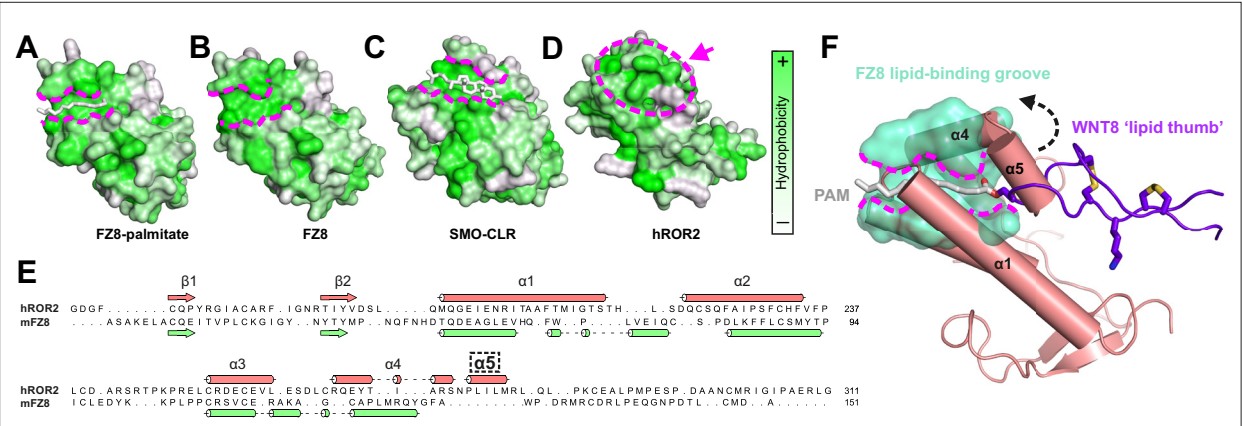

**Figure 2.** Comparison of the ROR2 cysteine-rich domain (CRD) to other Frizzled (FZ)-type CRDs. (**A–D**) CRDs are shown as surface representation and colored according to hydrophobicity (green: hydrophobic to white: hydrophilic). Displayed structures: (**A**) FZ8-PAM (palmitoleate) complex (PDB 4F0A) (*Janda et al., 2012*), (**B**) FZ8-apo (PDB 1IJY) (*Dann et al., 2001*), (**C**) Smoothened-CLR (cholesterol) complex (PDB 5L7D) (*Byrne et al., 2016*), and (**D**) ROR2 (from this study). (**E**) Structure-based sequence alignment of the ROR2 and FZ8 CRDs. The ROR2 inserted helix α5 is shown with a dashed box. (**F**) Structural superposition of the ROR2 CRD (salmon) with the FZ8-WNT8 (green/purple, PDB 4F0A) complex. Only the WNT8 'lipid thumb' is shown, with the covalently attached palmitoleate modification in white.

The online version of this article includes the following figure supplement(s) for figure 2:

**Figure supplement 1.** Small-angle X-ray scattering (SAXS) solution structure of the ROR2 extracellular region (ECD).

**Figure supplement 2.** Comparison of the human ROR2 and *Drosophila* Nrk cysteine-rich domains (CRDs).

domain could mediate dimerization. Structurally, the ROR2 CRD is evolutionarily related to other FZ-type 'groove-containing' CRDs (*Nachtergaele et al., 2013*), such as MuSK and FZ8 (*Figure 1D* and *Supplementary file 2*; *Stiegler et al., 2009*; *Dann et al., 2001*; *Janda et al., 2012*), as well as the cholesterol-binding Hedgehog signal transducer Smo (*Byrne et al., 2016*). These are structurally distinct to the 'pocket-type' CRDs such as NPC1 and RFBP, which bury their physiological ligands in deep cavities (*Bazan and de Sauvage, 2009*).

## Structural analysis of ROR2 CRD function

The FZ-type CRDs from both FZ8 and Smo exhibit shallow hydrophobic grooves for the binding of palmitoleate or cholesterol, respectively (*Janda et al., 2012*; *Byrne et al., 2016*; *Figure 2A–C*). One general characteristic differentiating this subfamily of CRDs from the 'pocket-type' subfamily is that grooves are structurally conserved when ligand free (*Figure 2A and B*), with minimal structural rearrangement occurring upon ligand binding (*Dann et al., 2001*; *Janda et al., 2012*; *Nachtergaele et al., 2013*; *Byrne et al., 2016*). Despite structural conservation with other FZ-type receptors, the ROR2 CRD does not contain a visible hydrophobic groove pre-formed for ligand recognition (*Figure 2D*). A structure-based sequence alignment shows that the ROR2 CRD has evolved an additional helical insertion (α5) relative to the FZ8 CRD (*Figure 2E*). Structural superposition of the ROR2 CRD with the FZ8:WNT-palmitoleate binary complex (*Janda et al., 2012*) shows that this helical insertion blocks exposure of any possible palmitoleate-binding groove (*Figure 2F*, *Figure 1—figure supplement 1F–H*). This observation is therefore incompatible with a direct binding event occurring between the ROR2 CRD and the WNT5A palmitoleate moiety, suggesting that the high affinity 'site 1' WNT5A interaction must occur either via a different site on the CRD or through a separate co-receptor(s), or require structural rearrangements as-yet not observed for groove-containing FZ-CRDs.

The structure of the Nrk CRD revealed a deeply buried fatty acid (*Shi et al., 2021*), whereas no bound lipid, nor an internal space that could potentially accommodate it, was observed in our human ROR2 CRD structure (*Figure 2D and F*). Structural comparison of the ROR2 and Nrk CRDs showed that the region of Nrk that interacts with the lipid does not superpose well with the analogous region in ROR2 (*Figure 2—figure supplement 2*). Notably, residue K170 of Nrk that interacts with the head group of the fatty acid is absent in ROR2 (*Figure 1—figure supplement 2B*). Another basic side chain that contacts the head group of the lipid in Nrk is R179 – this residue is conserved in ROR2 (R280), but its side chain protrudes into solution (*Figure 2—figure supplement 2*). It is noteworthy that in FZ-CRD structures for which apo- and lipid-bound structures are available (e.g. FZ4), major structural rearrangements within the CRD are not observed upon ligand recognition (*Shen et al., 2015*; *DeBruine et al., 2017*). Taken together, these observations suggest that ROR2 is unlikely to bind an internally buried free fatty acid analogous to that observed in Nrk.

## Functional requirement of the ROR2 CRD in WNT5A signaling

We next examined the requirement of the ROR2 CRD, as well as that of other ROR2 domains, in WNT5A signaling. We first developed a central rescue paradigm that allowed us to exogenously express various ROR2 mutant proteins in *Ror1/Ror2* double knockout (ROR KO) MEFs and assess their ability to restore WNT5A signaling (*Figure 3A*). For these experiments, we isolated primary MEFs from E12.5 *Ror1^f/f*; *Ror2^f/f*; *CAG-CreER* embryos that carry conditional (floxed, or *f*) alleles of *Ror1* and *Ror2*, as well as a *CreER* transgene driven by the ubiquitously expressed *CAG* promoter (*Susman et al., 2017*; *Hayashi and McMahon, 2002*). E12.5 was chosen because we previously showed that MEFs from this embryonic age are particularly responsive to WNT5A-ROR signaling (*Susman et al., 2017*; *Konopelski Snavely et al., 2021*). To enable long-term genetic manipulations, we immortalized the MEFs (called iMEFs) via Cas9/CRISPR-mediated ablation of the *Tp53* gene (*Dirac and Bernards, 2003*). To allow quantitative measurement of WNT5A-ROR signaling, we further engineered a GFP-Pdzrn3 degradation reporter construct into the iMEFs. We previously reported that activation of WNT5A-ROR signaling induces the proteasomal degradation of the downstream effector protein Pdzrn3, and that this regulation could be quantified using the GFP-Pdzrn3 reporter in live cells via flow cytometry (*Konopelski Snavely et al., 2021*). We next treated the iMEF reporter cells with 4-hydroxytamoxifen (4OHT), which activates the genetically encoded CreER protein to induce excision of the floxed *Ror1* and *Ror2* alleles. Lastly, to compare the activity of wildtype (WT) ROR2 versus its mutant derivatives, we developed a lentivirus-based gene replacement strategy that allowed

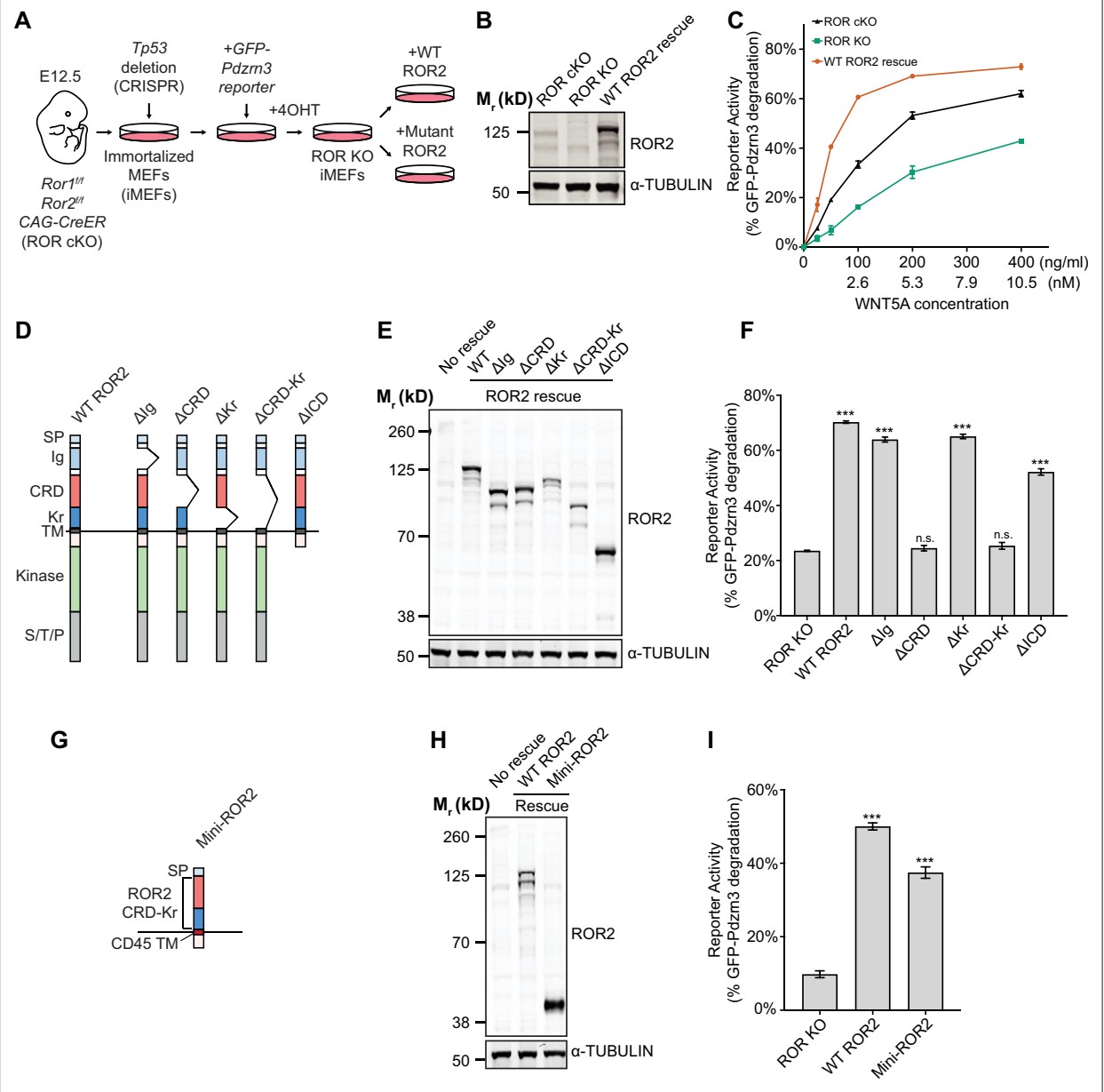

**Figure 3.** Requirement of the ROR2 cysteine-rich domain (CRD) in WNT5A signaling. (**A**) Workflow of the ROR2 central rescue paradigm. Primary mouse embryonic fibroblasts (MEFs) generated from E12.5 *Ror1^{f/f}; Ror2^{f/f}; CAG-CreER* conditional knockout (ROR cKO) mouse embryos were immortalized by CRISPR/Cas9-mediated deletion of the *Tp53* gene. A WNT5A-ROR signaling reporter (GFP-Pdzrn3) was stably inserted in the immortalized MEFs (iMEFs) via piggyBac-based transposition. ROR cKO iMEFs were then treated with 4-hydroxytamoxifen (4OHT) to activate the CreER recombinase to delete the *Ror1* and *Ror2* genes, resulting in ROR knockout (ROR KO) iMEFs (genotype: *Ror1^{-/-}; Ror2^{-/-}; CAG-CreER*). To compare the function of ROR2 domain truncation mutants, wildtype (WT) or mutant ROR2 rescue constructs were re-expressed in ROR KO iMEFs via lentiviral transduction. (**B**) Western blot showing the expression of endogenous ROR2 (the 125kD band) in ROR cKO iMEFs, the loss of ROR2 expression in ROR KO iMEFs, and re-expression of the WT ROR2 rescue construct (Flag-tagged). (**C**) Dose-response curves showing WNT5A-ROR signaling activity, assayed by GFP-Pdzrn3 degradation, as a function of WNT5A concentration in ROR cKO iMEFs, ROR KO iMEFs, or ROR KO iMEFs expressing the WT ROR2 rescue construct. All iMEFs were pretreated with Wnt-C59, an inhibitor of the membrane-bound *O*-acyltransferase Porcupine (***Proffitt et al., 2013***), to block the activity of endogenous WNTs. Each data point was calculated from the median fluorescence ([before WNT5A stimulation – after WNT5A stimulation]/before WNT5A stimulation) of the GFP-Pdzrn3 reporter from 10,000 cells. Error bars represent ± SEM calculated from two technical replicates (two independent WNT5A stimulation experiments of the same cell lines). (**D**) Schematic of ROR2 domain truncation mutants. (**E**) Western blot showing the expression of the WT and mutant ROR2 constructs. ROR2 protein variants were detected using a rabbit polyclonal anti-ROR2 antibody. α-TUBULIN was used as the loading control. (**F**) Quantification of the effects of ROR2 mutant variants in rescuing WNT5A-ROR signaling, as assayed by GFP-Pdzrn3 degradation. Cells were treated with 200 ng/ml (5.3 nM) WNT5A for 6 hr. Error bars represent ± SEM calculated from three technical replicates. t-Test (unpaired) was performed to determine statistical significance of each rescue construct vs. the no rescue control (ROR KO cells). (**G**) Schematic of the mini-ROR2

*Figure 3 continued on next page*

*Figure 3 continued*

construct. (**H**) Anti-ROR2 western blot showing the expression of the WT ROR2 and mini-ROR2 rescue constructs. α-TUBULIN was used as the loading control. (**I**) Quantification of the effect of mini-ROR2 in rescuing WNT5A-ROR signaling, as assayed by GFP-Pdzrn3 degradation. Cells were treated with 200 ng/ml (5.3 nM) WNT5A for 6 hr. Error bars represent ± SEM calculated from three technical replicates. t-Test (unpaired) was performed to determine statistical significance of each rescue construct vs. the no rescue control (ROR KO cells). *Figure 3—source data 1* (related to panel C). *Figure 3—source data 2* (related to panel F). *Figure 3—source data 3* (related to panel I).

The online version of this article includes the following source data and figure supplement(s) for figure 3:

**Source data 1.** Dose-response curves showing WNT5A-ROR signaling activity, as assayed by GFP-Pdzrn3 degradation (related to panel C).

**Source data 2.** Quantification of the effects of ROR2 mutant variants in rescuing WNT5A-ROR signaling, as assayed by GFP-Pdzrn3 degradation (related to panel F).

**Source data 3.** Quantification of the effect of mini-ROR2 in rescuing WNT5A-ROR signaling, as assayed by GFP-Pdzrn3 degradation (related to panel I).

**Figure supplement 1.** Surface expression analysis of ROR2 domain truncation mutants and mini-ROR2.

**Figure supplement 2.** Requirement of the ROR2 cysteine-rich domain (CRD) in WNT5A signaling, as assayed by Dishevelled 2 (DVL2) phosphorylation.

**Figure supplement 3.** The ability of mini-ROR2 to rescue WNT5A signaling, as assayed by Dishevelled 2 (DVL2) phosphorylation.

the stable expression of ROR2 'rescue' constructs (*Figure 3A–C*). WNT5A dose-response analysis comparing ROR conditional KO (cKO) reporter cells versus ROR KO cells (i.e. cKO cells treated with 4OHT) showed that loss of ROR1 and ROR2 expression substantially decreased WNT5A signaling across all WNT5a doses tested, and that this deficit can be fully rescued by the expression of an exogenous WT ROR2 construct (*Figure 3C*). Interestingly, ROR KO reporter cells without any ROR2 rescue still retained some basal WNT5A signaling activity, which remained dose-dependent with respect to WNT5A concentration (*Figure 3C*). This observation indicates that, while ROR receptors play an important role in promoting WNT5A signaling, additional receptor(s) exist in these cells to transmit the WNT5A signal.

To systematically identify the domain(s) of ROR2 required for WNT5A signaling, we generated a series of ROR2 domain truncation mutants based on the folded domain boundaries identified by our crystal structure (*Figure 3D*) and assessed their ability to restore WNT5A signaling in the iMEF signaling rescue assay. Immunoblotting confirmed that the mutant proteins were expressed at levels comparable to the WT ROR2 rescue construct (*Figure 3E*). Cell surface staining using the Flag epitope tag fused to the N-terminus of the WT and mutant ROR2 rescue constructs further confirmed that these constructs are expressed on the plasma membrane at comparable levels (*Figure 3—figure supplement 1A*). Signaling analysis using the GFP-Pdzrn3 degradation reporter revealed that the two ROR2 mutants lacking the CRD (ΔCRD and ΔCRD-Kr) completely failed to restore WNT5A-induced degradation of the GFP-Pdzrn3 reporter (*Figure 3F*). The same defects were observed when we assayed DVL2 phosphorylation as an independent readout of WNT5A-ROR signaling (*Ho et al., 2012*; *Figure 3—figure supplement 2A and B*). These results thus established that the CRD is essential for WNT5A signaling. Surprisingly, all other mutants in the series, including one lacking almost the entire intracellular region (ΔICD), still retain significant signaling capability, indicating that only the CRD is indispensable for the core function of ROR2 in promoting WNT5A signaling (*Figure 3F*, *Figure 3—figure supplement 2A and B*).

We next assessed the sufficiency of the ROR2 CRD in mediating WNT5A signaling. We engineered a chimeric construct (mini-ROR2) in which the isolated ROR2 CRD and Kr tandem domains are fused to a generic TM helix derived from the unrelated protein CD45 (*Chin et al., 2005*), followed by a small, intracellular juxtamembrane fragment (*Figure 3G*). Immunoblotting and cell surface staining experiments confirmed the protein and cell surface expression of mini-ROR2 (*Figure 3H* and *Figure 3—figure supplement 1B*). Signaling analyses by GFP-Pdzrn3 reporter degradation and DVL2 phosphorylation showed that mini-ROR2 can substantially rescue WNT5A signaling (*Figure 3I*, *Figure 3—figure supplement 3A and B*). This experiment, taken together with the truncation analysis, established that the ROR2 CRD is required and minimally sufficient to support the function of ROR2 in WNT5A signaling.

## Role of FZ receptors in WNT5A-ROR signaling

Based on our observations that (1) the ROR2 CRD lacks the hydrophobic groove that binds to the lipid modification of WNT5A (*Figure 2D and F*), (2) iMEFs lacking both ROR1 and ROR2 still retain some

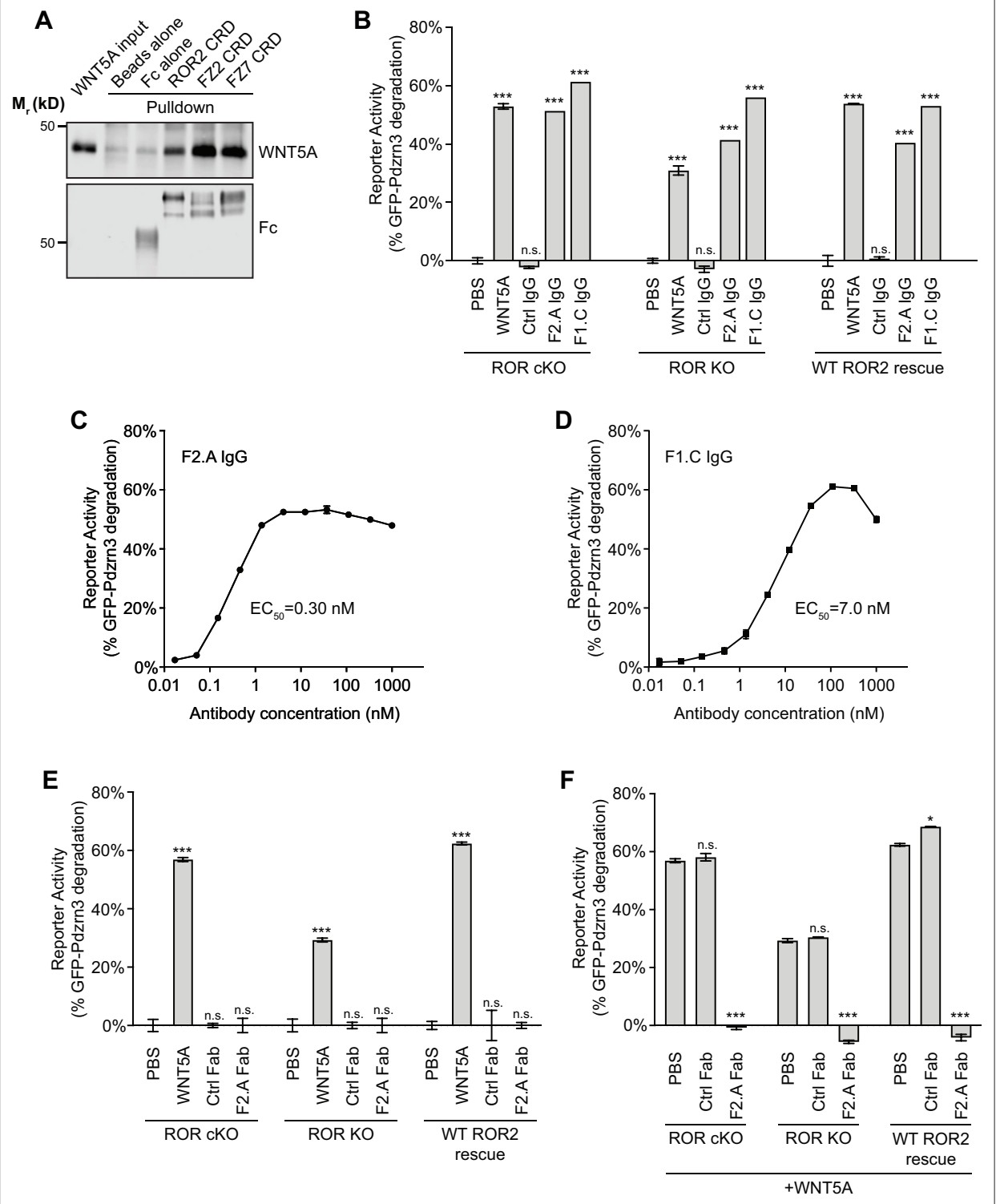

**Figure 4.** Involvement of Frizzled (FZ) in WNT5A-ROR signaling. (**A**) Fc fusions of the ROR2, FZ2, and FZ7 cysteine-rich domains (CRDs) immobilized on protein A-coated beads were tested for their ability to pull down WNT5A. Protein A beads alone and protein A beads coated with Fc were used as negative controls. 9.3% of the input and 33% of the pulled down materials were analyzed on a 12% SDS-polyacrylamide gel. WNT5A and Fc fusion proteins were detected by western blotting using anti-WNT5A and anti-Flag antibodies (all Fc fusions were tagged with the Flag epitope), respectively. (**B**) Quantification of the effects of the pan-anti-FZ IgG F2.A and anti-FZ1 IgG F1.C in inducing GFP-Pdzrn3 reporter degradation in ROR conditional knockout (cKO) immortalized mouse embryonic fibroblasts (iMEFs), ROR KO iMEFs, and ROR KO iMEFs rescued with wildtype (WT) ROR2. WNT5A was used as the positive control, and an isotype-matching IgG against the Gaussia luciferase was used as the negative control (Ctrl IgG). No IgGs and

*Figure 4 continued on next page*

*Figure 4 continued*

WNT5A were mixed in this experiment. All IgGs were used at 200 nM, and WNT5A was used at 200 ng/ml (5.3 nM). WNT5A and FZ IgG stimulations were done in the presence of Wnt-C59 for 6 hr. Each data point was calculated from the median fluorescence ([before antibody treatment – after antibody treatment]/before antibody treatment) of the GFP-Pdzrn3 reporter. Error bars represent ± SEM calculated from three technical replicates; 20,000 cells per replicate. t-Test (unpaired) was performed to determine the statistical significance of each treatment vs. PBS. (C) Dose-response analysis showing the ability of the bivalent pan-anti-FZ IgG F2.A to activate WNT5A signaling, as assayed by GFP-Pdzrn3 degradation in ROR KO iMEFs treated with Wnt-C59 but without WNT5A. (D) Dose-response analysis showing the ability of the bivalent anti-FZ1 IgG F1.C to activate WNT5A signaling, as assayed by GFP-Pdzrn3 degradation in ROR KO iMEFs treated with Wnt-C59 but without WNT5A. (E) Quantification of the effects of the monovalent Fab fragment of the F2.A antibody in inducing GFP-Pdzrn3 degradation in ROR cKO iMEFs, ROR KO iMEFs, and ROR KO iMEFs rescued with WT ROR2. All iMEFs were treated with Wnt-C59. WNT5A was used as the positive control, and an isotype-matching Fab against the Gaussia luciferase was used as the negative control (Ctrl Fab). The Ctrl and F2.A Fabs were used at 200 nM, and WNT5A was used at 200 ng/ml (5.3 nM). No cells were simultaneously treated with Fab and WNT5A in this experiment. Each data point was calculated from the median fluorescence ([before antibody treatment – after antibody treatment]/before antibody treatment) of the GFP-Pdzrn3 reporter. Error bars represent ± SEM calculated from three technical replicates; 20,000 cells per replicate. t-Test (unpaired) was performed to determine statistical significance of each treatment vs. PBS. (F) Quantification of the effects of the monovalent Fab fragment of the F2.A antibody in inhibiting WNT5A-induced GFP-Pdzrn3 reporter degradation in ROR cKO iMEFs, ROR KO iMEFs, and ROR KO iMEFs rescued with WT ROR2. All iMEFs were treated with Wnt-C59. An isotype-matching Fab against the Gaussia luciferase was used at the negative control (Ctrl Fab). In all conditions, cells were pretreated with PBS, Ctrl Fab, or F2.A Fab for 30 min and then stimulated with WNT5A in the presence of PBS or Fabs for an additional 6 hr. The Ctrl and F2.A Fabs were used at 200 nM, and WNT5A was used at 200 μg/ml (5.3 nM). Each data point was calculated from the median fluorescence ([before WNT5A treatment – after WNT5A treatment]/before WNT5A treatment) of the GFP-Pdzrn3 reporter. Error bars represent ± SEM calculated from three technical replicates; 20,000 cells per replicate. t-Test (unpaired) was performed to determine statistical significance of each treatment vs. PBS. *Figure 4—source data 1* (related to panel B). *Figure 4—source data 2* (related to panel C). *Figure 4—source data 3* (related to panel D). *Figure 4—source data 4* (related to panel E). *Figure 4—source data 5* (related to panel F).

The online version of this article includes the following source data and figure supplement(s) for figure 4:

**Source data 1.** Quantification of the effects of the anti-pan Frizzled (FZ) IgG F2.A and anti-FZ1 IgG F1.C in inducing GFP-Pdzrn3 reporter degradation (related to panel B).

**Source data 2.** Dose-response analysis showing the ability of the bivalent anti-pan Frizzled (FZ) IgG F2.A to activate WNT5A signaling, as assayed by GFP-Pdzrn3 degradation (related to panel C).

**Source data 3.** Dose-response analysis showing the ability of the bivalent anti-FZ1 IgG F1.C to activate WNT5A signaling, as assayed by GFP-Pdzrn3 degradation (related to panel D).

**Source data 4.** Quantification of the effects of the monovalent Fab fragment of the F2.A antibody in inducing GFP-Pdzrn3 reporter degradation (related to panel E).

**Source data 5.** Quantification of the effects of the monovalent Fab fragment of the F2.A antibody in inhibiting WNT5A-induced GFP-Pdzrn3 reporter degradation (related to panel F).

**Figure supplement 1.** Activation of WNT5A signaling by anti-Frizzled (FZ) IgGs, as assayed by Dishevelled 2 (DVL2) phosphorylation.

signaling activity (*Figure 3C*), and (3) ROR2 lacking the intracellular region can still signal (*Figure 3F*), we postulated that ROR2 cannot by itself function as the signaling receptor for WNT5A; instead, it likely facilitates the signaling function of another receptor(s). Members of the FZ family, particularly the FZ1, FZ2, and FZ7 subfamily, are plausible candidates, as numerous previous studies have implicated them in aspects of noncanonical WNT5A signaling (*Oishi et al., 2003*; *Bryja et al., 2007*; *Sato et al., 2010*; *Grumolato et al., 2010*; *Yu et al., 2012*; *Konopelski Snavely et al., 2021*). Furthermore, a human mutation in FZ2 was recently identified in RS patients (*White et al., 2018*). To explore the role of the FZ family in WNT5A-ROR2 signaling, we first compared the ability of the ROR2, FZ2, and FZ7 CRDs to bind WNT5A. We expressed and purified these CRDs as secreted Ig Fc domain fusion constructs from HEK293T cells, followed by in vitro pulldown assays using purified recombinant WNT5A. The experiment showed that WNT5A can interact with all three CRDs tested, and that the binding affinity appears to be higher for the FZ2 and FZ7 CRDs than for the ROR2 CRD (*Figure 4A*). This finding is consistent with our structural data and with previous binding studies (*Oishi et al., 2003*; *Mikels and Nusse, 2006*; *Sato et al., 2010*), and suggests that despite lacking the hydrophobic groove, the ROR2 CRD can still bind WNT5A via a presumably lipid-independent manner.

To further test whether the binding interaction between WNT5A and the FZ CRD is important for signaling, we treated iMEF reporter cells with a synthetic monoclonal antibody, F2.A, that binds the CRD of six FZ family members, including FZ1, FZ2, and FZ7, and competes with WNTs for FZ binding (*Pavlovic et al., 2018*). We hypothesized that if FZs participate in WNT5A-ROR2 signaling, then treatment with this antibody should block the ability of WNT5A to induce signaling. Surprisingly, we found that F2.A, in its bivalent IgG form and in the absence of any WNT5A,

strongly induced GFP-Pdzrn3 reporter degradation, as well as DVL2 phosphorylation, in ROR cKO iMEFs, ROR KO iMEFs, and WT ROR2 rescued iMEFs (*Figure 4B*; *Figure 4—figure supplement 1*). An independent bivalent IgG F1.C, which specifically targets the CRD of FZ1, similarly induced GFP-Pdzrn3 degradation and DVL2 phosphorylation (*Figure 4B*; *Figure 4—figure supplement 1*). Dose-response analyses of the F2.A and F1.C IgGs showed that these antibodies are highly potent in inducing Pdzrn3 degradation, with the respective EC$_{50}$ values of 0.30 nM and 7.0 nM (*Figure 4C and D*).

Moreover, we observed that at post-saturating concentrations, both IgGs exhibited reduced activity (*Figure 4C and D*), suggesting that they might function by inducing FZ dimerization. To further explore this idea and the functional role of FZs in WNT5A signaling, we tested the monovalent antigen-binding fragment (Fab) of F2.A and found that it not only lost the ability to induce signaling on its own (*Figure 4E*), possibly because it could no longer dimerize FZs, but instead, acted as a potent antagonist of WNT5A-dependent pathway activation (*Figure 4F*), presumably by blocking the WNT5A-FZ CRD interaction. Altogether, these data support a model in which FZs function as the signaling receptors for WNT5A, and ROR2 acts through its CRD to enhance FZ function, possibly by promoting WNT5A-FZ interactions and/or FZ dimerization (Figure 6; see Discussion).

## RS mutations in the ROR2 CRD and Kr domains compromise ROR2 trafficking to the cell surface

Of all the ROR2 domains, the CRD and Kr domains are most frequently mutated in RS patients (*Tufan et al., 2005*; *Afzal et al., 2000*; *Tamhankar et al., 2014*; *Mehawej et al., 2012*). Our structural and functional data suggest that these mutations would disrupt the function of ROR2 in WNT5A signaling. To test this hypothesis, we expressed and characterized five substitution mutations from Robinow patients that map to the CRD (C182Y, R184C, R189W, C223Y, R272C) and two that map to the Kr domain (G326A and R366W), using the ROR KO iMEF rescue system. Western analysis showed that all seven mutant proteins were expressed at comparable levels as WT ROR2 (*Figure 5A*). In WNT5A signaling assays, we found that three of the five CRD mutations (C182Y, R184C, and C223Y) and both Kr mutations (G326A and R366W) exhibited strongly reduced signaling capabilities (*Figure 5B*, *Figure 5—figure supplement 1*). The remaining two mutants (R189W and R272C) exhibited milder but still statistically significant signaling deficits (*Figure 5B*, *Figure 5—figure supplement 1*).

We noticed that the five mutants with severe signaling defects (C182Y, R184C, C223Y, G326A, and R366W) all exhibited increased gel mobility in western analysis (*Figure 5A*), suggestive of possible defects in glycosylation/trafficking through the secretory pathway. Correspondingly, the two mutants with milder signaling defects (R189W and R272C) showed a partial increase in their gel mobility (*Figure 5A*). To assess the glycosylation state of the mutant constructs, we conducted endoglycosidase H (Endo H) and peptide-*N*-glycosidase F (PNGase F) sensitivity assays. Complex glycan modifications attached in the Golgi are resistant to Endo H, whereas both pre- and post-Golgi glycan modifications are generally sensitive to PNGase F. This analysis shows that C182Y, R184C, C223Y, G326A, and R366W are fully sensitive to both Endo H and PNGase F, indicating that these mutants fail to traffic to and beyond the Golgi (*Figure 5C*). In contrast, R189W and R272C are partially resistant to Endo H but fully sensitive to PNGase F, indicating that some of these proteins can traffic to and past the Golgi (*Figure 5C*). Lastly, we conducted surface staining experiments using the Flag tag fused to the N-terminus of the mutant constructs and found that C182Y, R184C, C223Y, G326A, and R366W all fail to traffic to the plasma membrane, whereas R189W and R272C are expressed on the cell surface (*Figure 5—figure supplement 2*).

Further insights into the pathogenic mechanisms of Robinow mutations were obtained from our structural analysis. C182Y and C223Y disrupt conserved cysteines in the CRD. Since our structure showed that all 10 cysteines in the CRD are involved in disulfide bond formation (*Figure 1B*), these mutations likely cause RS by destabilizing the core structure of the CRD (*Figure 5D–F*). Two other mutations in the CRD (R184C and R272C) involve amino acid substitution to cysteines. Since both residues are solvent exposed (*Figure 5D–F*), they may form cryptic intramolecular disulfide bonds that disrupt the protein fold, or alternatively form intermolecular disulfide bonds that cause inappropriate dimerization or oligomerization that results in protein aggregation. Based on our signaling data (*Figure 5B*), R184C is likely more prone to these perturbations than R272C. R189W showed the least functional deficit in our assay system. It is possible that even a subtle signaling deficit is sufficient

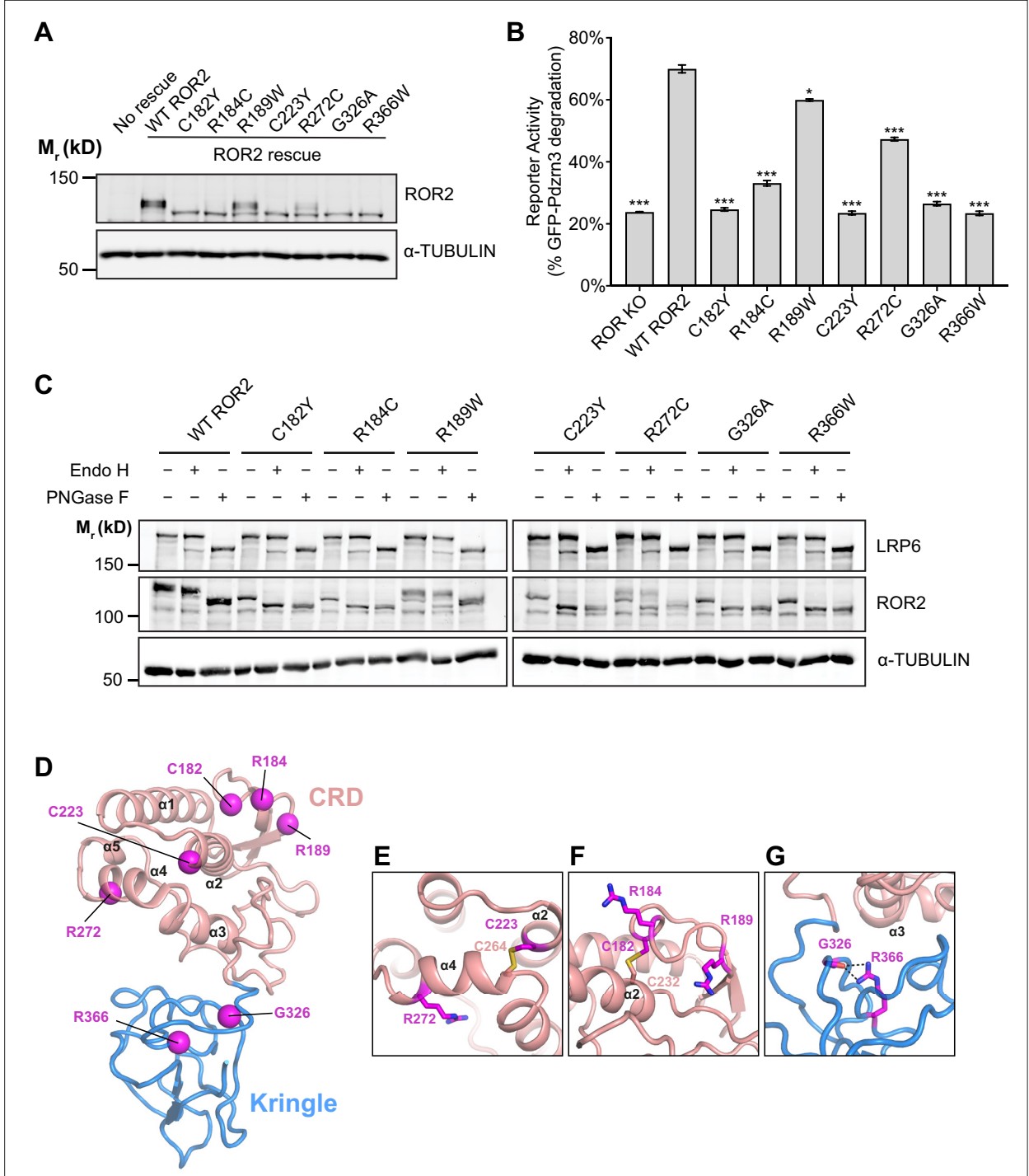

**Figure 5.** Analysis of Robinow syndrome mutations in the ROR2 cysteine-rich domain (CRD) and Kringle (Kr) domain. (**A**) Western blot showing expression of wildtype (WT) ROR2 and Robinow syndrome ROR2 mutants in the ROR knockout (KO) immortalized mouse embryonic fibroblast (iMEF) reporter cells. (**B**) Quantification of the effects of Robinow syndrome ROR2 mutants in rescuing WNT5A-ROR signaling, as assayed by GFP-Pdzrn3 degradation. Cells were treated with 200 ng/ml (5.3 nM) WNT5A for 6 hr. Error bars represent ± SEM calculated from three technical replicates. t-Test (unpaired) was performed to determine statistical significance for mutants vs. WT ROR2 rescue. (**C**) Western blots showing the sensitivity of WT ROR2 and Robinow syndrome ROR2 mutants to endoglycosidase H (Endo H) and peptide-*N*-glycosidase F (PNGase F). LRP6 was used as a control substrate for Endo H and PNGase F, and α-TUBULIN was used as the loading control. (**D**) Structure of the ROR2 CRD-Kr tandem domains showing the location of the Robinow syndrome mutations. (**E**) Close-up view of C223 and R272. (**F**) Close-up view of C182, R184, and R189. (**G**) Close-up view of G326 and R366.

*Figure 5—source data 1* (related to panel B).

*Figure 5 continued on next page*

*Figure 5 continued*

The online version of this article includes the following source data and figure supplement(s) for figure 5:

**Source data 1.** Quantification of the effects of Robinow syndrome ROR2 mutants in rescuing WNT5A-ROR signaling, as assayed by GFP-Pdzrn3 degradation (related to panel B).

**Figure supplement 1.** Signaling capabilities of Robinow syndrome mutants, as assayed by Dishevelled 2 (DVL2) phosphorylation.

**Figure supplement 2.** Surface expression analysis of Robinow syndrome mutations in the ROR2 cysteine-rich domain (CRD) and Kringle (Kr) domains.

to disrupt embryonic development and cause RS. Alternatively, R189W might be involved in other aspects of ROR2 regulation not readily detected in our assay system.

Interestingly, the two mutations that map to the Kr domain (G326A and R366W) are among the most detrimental (*Figure 5B*, *Figure 5—figure supplement 1*). Both of these mutations are located near the CRD-Kr interface (*Figure 5D and G*). G326 is situated near the linker between the CRD and Kr (*Figure 5D and G*), and therefore, substitution at this position may open up the space between the two domains and expose hydrophobic residues to promote protein aggregation, which in turn prevents trafficking out of the ER. Likewise, R366W may disrupt the overall fold of Kr, or disrupt the interface between the CRD and Kr to destabilize the CRD-Kr structural unit (*Figure 5D and G*). Together, these functional and structural analyses provide new insights into the molecular mechanisms of Robinow pathogenesis.

## Discussion

In this study, we used an integrated approach of structural biology, genetics, and pharmacology to better understand the mechanism of WNT5A signal reception at the cell surface. We made several key observations that substantially advance our current understanding of WNT5A receptor function.

First, by solving the crystal structure of the ROR2 CRD, we made the surprising finding that this domain lacks the characteristic hydrophobic groove that binds the acyl moiety of WNTs and is thus incompatible with high-affinity interaction with WNT ligands. Our binding assay showed that the binding affinity between WNT5A and the ROR2 CRD indeed appears to be weaker than that between WNT5A and FZ CRDs. These experimental data agree with the modeling work by Janda and Garcia, which predicted that the ROR2 CRD might not possess the hydrophobic groove to accommodate the

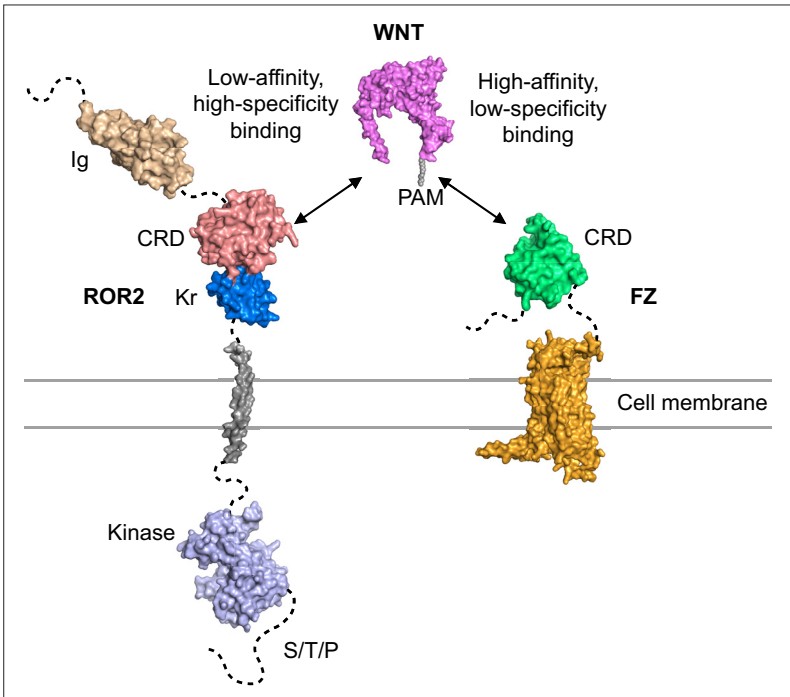

**Figure 6.** Model of ROR2 cysteine-rich domain (CRD) and Frizzled (FZ) action in WNT5A-ROR signaling.

lipid modification of WNTs (*Janda and Garcia, 2015*). This also agrees with the published crystal structure of MuSK, which is related to ROR2 and also lacks a hydrophobic groove in its CRD (*Stiegler et al., 2009*). The occlusion of the lipid/small molecule-binding site in ROR2 is unexpected and of general interest because this site was previously shown to play an important role not only for WNT-FZ binding and FZ dimerization during canonical WNT signaling, but also for Smo function during Hedgehog signaling (*Janda et al., 2012*; *Byrne et al., 2016*; *Nachtergaele et al., 2013*; *Nile et al., 2017*). As there is clear evidence that the mammalian WNT5A is lipidated (*Mikels and Nusse, 2006*), our data raised the question of which co-receptor(s) in the pathway, if not ROR, is (are) responsible for high-affinity WNT5A binding and signal transduction across the membrane. Though the exact identity of this co-receptor remains to be determined, our work points to the FZ receptor family, as experimental dimerization of FZ proteins using a highly specific bivalent IgG that binds the FZ CRDs is sufficient to mimic WNT5A signaling, whereas monovalent Fab of the same antibody inhibits the ability of WNT5A to initiate signaling. We therefore favor a model in which WNT5A interacts with FZ with high affinity to transduce signals across the plasma membrane, either by itself or in conjunction with another as-yet unidentified protein(s). The ROR2 CRD appears to possess a low-affinity WNT5A binding site, possibly analogous to the low-affinity site '2' observed in the WNT8-FZ8 complex (*Janda et al., 2012*). The ROR2 CRD could possibly potentiate WNT5A signaling by enhancing the binding interaction between WNT5A and FZ via the formation of an FZ:WNT:ROR2 super-complex (*Figure 6*). In this scenario, the WNT5A palmitoleate modification engages the FZ groove as previously described ('site 1') (*Janda et al., 2012*), while ROR2 binds at site 2 to further promote WNT5A recruitment to the receptor complex, and/or recruit intracellular effectors of noncanonical WNT signaling. This model further suggests that the high-affinity WNT5A palmitoleate-FZ interaction acts as an indiscriminate anchor to stabilize the overall receptor super-complex and initiate downstream signaling, and that the low-affinity WNT5A-ROR2 CRD interaction contributes to ligand-receptor specificity for noncanonical WNT5A signaling. Lastly, it is also possible that ROR2 CRD can act by inducing an allosteric change in the structure of FZ to enhance FZ function or promote FZ dimerization that in turn increases FZ's affinity for WNT5A (*Carron et al., 2003*; *Nile et al., 2017*).

Second, we observed that ROR2 mutants lacking the intracellular domain (ΔICD and mini-ROR2) can still support WNT5A signaling. This is in line with the idea that ROR2 itself is unlikely to be the signal-transmitting receptor for WNT5A, and that the cytoplasmic domain of ROR2 might have evolved to play a regulatory role, such as in receptor stability, signal amplification, and/or termination. This idea is further supported by the observation that some residual signaling activity remains in cells lacking both ROR1 and ROR2 when stimulated with exogenously added WNT5A, and that the structure of the ROR2 kinase domain predicts it to be catalytically inactive (*Artim et al., 2012*). Collectively, these findings firmly established a co-requirement for both ROR and FZ activities in noncanonical WNT5A signal transduction. This model is also consistent with previous in vivo work showing that *Ror1/Ror2* double knockout mice phenocopy the characteristic tissue truncation phenotypes of *Wnt5a* KO mice (*Nomi et al., 2001*; *Oishi et al., 2003*; *Ho et al., 2012*), and that human mutations in *WNT5A*, *ROR2*, or *FZ2* can all cause RS with similar developmental abnormalities (*Nagasaki et al., 2018*; *White et al., 2018*; *Person et al., 2010*; *Afzal et al., 2000*). Though several previous co-immunoprecipitation experiments have shown binding interactions between WNT5A and ROR2 (*Oishi et al., 2003*; *Mikels and Nusse, 2006*), between WNT5A and FZ family members (*Sato et al., 2010*), and between FZ and ROR proteins (*Oishi et al., 2003*), the biochemical details of these interactions remain to be characterized. In light of our present work, it is crucial to further define these interactions quantitatively in future studies and to understand their functions in the context of a co-receptor super-complex. Nonetheless, our results showing that ROR2 acts through its CRD to sensitize the function of FZs or an FZ-containing receptor complex during WNT5A-ROR signaling form the foundation for future studies.

Lastly, our work provides new insights into the molecular mechanisms of Robinow pathogenesis. Using the gene rescue strategy in iMEFs, we were able to directly assay the function of RS mutant variants under highly physiological conditions. We found that nearly all of the mutants tested exhibited defects in their trafficking to the cell surface and hence WNT5A signaling, and that mutating the cysteines required for disulfide bonds in the CRD was not tolerated. Furthermore, by combining these functional data with our structural analysis, we can classify the mutations based on their locations on the CRD-Kr structure and infer potential underlying mechanisms of structural and signaling perturbation. We envision that the experimental approach described in this study will serve as an important

model for interrogating other mutations in the pathway that cause RS, BDB, and cancer metastasis, and more generally, as a paradigm for modeling genetic disorders.

## Methods

### Protein expression and purification

Constructs of human ROR2 (GenBank ID: 19743898) comprising the ECD (residues 34–403), Ig-CRD (60–307), CRD-Kr (171–396), and CRD (171–307) were cloned into the pHLsec vector in frame with a C-terminal His$_6$-tag (*Aricescu et al., 2006*). ROR2 constructs were expressed by transient transfection in HEK293T cells with the addition of glycosylation inhibitor kifunensine (*Chang et al., 2007*). Proteins were isolated from dialyzed conditioned medium via immobilized metal-affinity chromatography and further purified via SEC in a buffer containing 10 mM HEPES pH 7.5, 150 mM NaCl.

### Crystallization and data collection

Prior to crystallization trials, ROR2 CRD-Kr was concentrated via ultrafiltration to a final concentration of 25 mg/ml and deglycosylated using catalytic quantities of endoglycosidase F1 (*Chang et al., 2007*) (0.2 µl/50 µl protein solution). Nanolitre-scale crystallization trials were performed using a Cartesian Technologies robot (100 nl protein plus 100 nl reservoir solution) in 96-well Greiner plates (*Walter et al., 2005*). ROR2 CRD-Kr crystallized in 0.1 M HEPES pH 7.5, 1.5 M LiSO$_4$ at a temperature of 25°C. Diffraction data were collected at a temperature of 100 K with crystals mounted within a liquid N2 cryo-stream. Crystals were treated with 20% (vol/vol) glycerol supplemented with reservoir solution and flash-cooled in liquid N2 prior to data collection. For Pt-SAD experiments, ROR2 CRD-Kr crystals were soaked in a saturated solution of K$_2$PtCl$_6$ made up in 0.1 M HEPES pH 7.5, 1.5 M LiSO$_4$ for 1 hr at 25°C, prior to cryoprotection and harvesting. Data were collected using the rotation method. Diffraction data were scaled and merged using the XIA2 suite and autoPROC (*Evans, 2006*; *Kabsch, 1988*; *Winter, 2010*; *Vonrhein et al., 2011*; *Tickle et al., 2018*).

### Structure solution

Initial phases for ROR2 CRD-Kr were obtained using Phenix Autosol with Pt-SAD data (*Terwilliger et al., 2009*). Four high occupancy Pt sites were identified from substructure solution, and automated model building of the resultant electron density map was performed using the program *Buccaneer* (*Cowtan, 2006*). This produced an interpretable model for the CRD (residues 174–307), but phases were not of a high enough quality to properly trace the Kr domain (residues 308–396). Subsequently, the CRD model generated was utilized as a molecular replacement search model in *Phaser* (*McCoy et al., 2007*) against higher resolution native data. This solution was fixed and a second search using a homology model for the Kr domain (generated via Swiss-Model) was performed (*Waterhouse et al., 2018*). This strategy resulted in higher scores in Phaser (LLG = 424, TFZ = 18.9) than searching for the CRD alone (LLG = 94, TFZ = 9.2), indicative of an improved solution. The model for the ROR2 CRD-Kr was manually built using COOT (*Emsley and Cowtan, 2004*) and refined to completion using Auto-BUSTER (*Smart et al., 2012*) and Phenix (*Terwilliger et al., 2009*). Data collection and refinement statistics are shown in *Supplementary file 1*.

### Structure analysis

Stereochemistry was assessed using the MolProbity server (*Davis et al., 2007*). Superpositions were calculated using *Pymol* (*Schrodinger, 2015*), which was also used to create ray-traced protein structure images for figures. Residues involved in interactions were identified using both the PDBsum and Pisa servers (*Laskowski, 2001*; *Krissinel and Henrick, 2007*). The solvent accessible radius was set to 1.4 Å for the representation of all protein surfaces. Evolutionary structural analysis of CRDs was performed with *SHP* (*Stuart et al., 1979*; *Riffel et al., 2002*) and *PHYLIP* (*Felsenstein, 1989*) to assemble a phylogenetic tree. The structure-based sequence alignment of ROR2 were generated using *UCSF Chimera* (*Pettersen et al., 2004*) and prepared for presentation using *ALINE* (*Okabayashi et al., 1991*).

## SEC-multiangle light scattering

SEC-multiangle light scattering (SEC-MALS) experiments were performed using an S200 10/30 column (GE Healthcare) equilibrated in a running buffer of 10 mM HEPES pH 7.5, 150 mM NaCl and coupled to a Wyatt Dawn HELEOS-II MALS detector and Wyatt Optilab rEX refractive index monitor. A 100 µl sample of purified ROR2 ECD was injected at a concentration of 48 µM. ASTRA software (Wyatt Technology) was utilized for data analysis.

## Small-angle X-ray scattering

SAXS experiments were carried out on beamline B21, Diamond Light Source, UK at 25°C, over a momentum transfer (q) range of 0.01 Å$^{-1}$ < q < 0.45 Å$^{-1}$, where q=4$\pi$sin($\theta$)/$\lambda$, and 2$\theta$ is the scattering angle. ROR2 CRD-Kr and the ROR2 ECD were injected onto an in-line Shodex KW-402.5 SEC column at concentrations of 6.5 mg/ml and 8 mg/ml respectively, both in a running buffer of 10 mM Tris pH 7.5, 150 mM NaCl, 1 mM KNO$_3$. Data were collected with a beam energy of 12.4 keV using a Pilatus P3-2M detector. Data processing and reduction was performed using the program Scatter. Missing residues were added using Modeller (*Eswar et al., 2003*) and all-atom models generated using Allosmod (*Weinkam et al., 2012*). A model for the ROR2 Ig domain was generated using the HHpred server (*Soding et al., 2005*). In each case 50 independent ensembles of 100 models were created. Calculation and fitting of theoretical scattering curves to collected data was performed by FoXS (*Schneidman-Duhovny et al., 2010*). This procedure was automated via the use of Allosmod-FoXS (*Guttman et al., 2013*). The best model from Allosmod-FoXS was then used as input for Multi-FoXS (*Schneidman-Duhovny et al., 2016*), for which flexible linker sequences were specified and 10,000 independent models calculated. Sugars were also modeled and simulated via this process. Theoretical SAXS curves were then calculated for each of these models and goodness-of-fit to the experimental data calculated using a chi-square value. Multi-state enumeration was then performed given the chi-square values using the 'branch-and-bound' method, to iteratively find the best multi-state model explaining the data. In the case of ROR2 ECD, MultiFoXS was necessary to find a two-state model which best explained the data based on flexibility in the linker between the Ig and CRD domains. For the ROR2 CRD-Kr construct, a single-state model calculated using Allosmod-FoXS was sufficient to account for the observed SAXS data.

## Mice

The *Ror1$^{f/f}$; Ror2$^{f/f}$; CAG-CreER* strain was previously described (*Ho et al., 2012*; *Susman et al., 2017*; *Hayashi and McMahon, 2002*) and maintained in a mixed 129 and C57BL/6J background. All animals were used according to the institutional and NIH guidelines approved by the Institutional Animal Care and Use Committee at University of California, Davis.

## Cell lines

HEK293T (CRL-3216, ATCC, Manassas, VA, USA) cells were initially purchased, re-authenticated by STR profiling (135-XV, ATCC), and tested negative for mycoplasma using the Universal Mycoplasma Detection Kit (30-1012K, ATCC). All cell lines were cultured at 37°C and 5% CO$_2$ in Dulbecco's Modified Eagle Medium (MT15017CV, Corning) supplemented with 1× glutamine (25-005-CI, Corning), 1× penicillin-streptomycin (30-002 CI, Corning) and 10% fetal bovine serum (16000069, Thermo Fisher Scientific).

For derivation of iMEF reporter cells, primary *Ror1$^{f/f}$; Ror2$^{f/f}$; CAG-CreER* MEFs were isolated directly from E12.5 mouse embryos as described (*Susman et al., 2017*). Passage 1 or 2 cultures were then immortalized by electroporating with Cas9/CRISPR constructs targeting the *Tp53* gene (Addgene Plasmids 88846 and 88847, gifts from Joan Massague *Wang et al., 2017*) using the Neon Transfection System (Thermo Fisher). Transformants were serially passaged for three to five generations until cells from the untransfected control group had died off. For 4OHT (H7904, Sigma-Aldrich) treatments, cells were treated with 0.25 µM of 4OHT on the first day and then 0.1 µM of 4OHT on the subsequent 3 days. The 4OHT containing media were replenished daily. To introduce the GFP-Pdzrn3 degradation reporter, a PB (PiggyBac)-GFP-Pdzrn3 plasmid, along with a Super PiggyBac transposase-expressing plasmid, was electroporated into *Ror1$^{f/f}$; Ror2$^{f/f}$; CAG-CreER* iMEFs and then cultured for 7 days. GFP-positive cells were sorted (MoFlo Astrios Cell Sorter, Beckman Coulter, 488 nm laser) to collect the weakly fluorescent (~lowest 1/3 among the GFP+ population on the FL scale) cells. The iMEF reporter

cells were authenticated by STR profiling (137-XV, ATCC), and tested negative for mycoplasma using the Universal Mycoplasma Detection Kit (30-1012K, ATCC).

### DNA constructs for rescue experiments

The full-length mouse *Ror2* open reading frame (ORF) was amplified from MEF cDNA and subsequently subcloned into a modified pENTR-2B vector carrying the signal sequence from human IgG and the Flag epitope tag upstream of the *Ror2* ORF; the native signal sequence of ROR2 was removed before subcloning. ROR2 truncation mutants and RS mutants were generated by Gibson assembly (*Gibson et al., 2009*). The following amino acid residues of the mouse ROR2 sequence were deleted in the truncation mutant series: ΔIg, 34–170; ΔCRD, 171–310; ΔKr, 313–402; ΔCRD-Kr, 171–402; ΔICD, 474–944. Mini-ROR2 consists of the IgG signal sequence, the Flag tag, CRD-Kr (amino acids 171–402), the monomeric version of the CD45 TM helix (*Chin et al., 2005*), and the ROR2 juxtamembrane sequence (amino acids 427–473) which contains the epitope for the polyclonal ROR2 antibody used for western detection. ORFs in the pENTR-2B donor vector were then transferred using the Gateway LR Clonase II enzyme mix (11791020, Thermo Fisher) into a modified pLEX_307 lentiviral vector (short EF1 pLEX_307) in which we removed the intron in the EF1 promoter to reduce transgene expression. The original pLEX_307 is a gift from David Root (Plasmid 41392, Addgene). The ORFs in all constructs were verified by Sanger sequencing.

### Lentiviral protein overexpression

Lentiviruses were packaged and produced in HEK293T cells by co-transfection of the lentiviral vectors with the following packaging plasmids: pRSV-REV, pMD-2-G, and pMD-Lg1-pRRE (gifts from Thomas Vierbuchen). 0.1 ml, 0.5 ml, or 2.5 ml of the viral supernatants was used to infect *Ror1^f/f^*; *Ror2^f/f^*; *CAG-CreER* iMEFs seeded at 50% confluency in 12-well plates for ~16 hr. Following removal of the virus-containing media, cells were cultured for 24 hr. Infected cells were then split and selected with puromycin (0.0015 mg/ml) for 4–5 days. Cells from the viral titer that killed a large proportion of cells (60–90%) were expanded and used for WNT5A signaling analysis; this ensured that the multiplicity of infection is ~1 for all cell lines used in the experiments, with the exception of mini-Ror2. The mini-Ror2 construct is more difficult to express and required a higher titer of virus (2.5 ml), as well as a higher concentration of puromycin (0.002 mg/ml).

### Antibodies

Antibodies against ROR2 were described previously (*Ho et al., 2012*; *Susman et al., 2017*) and used at 1 µg/ml for immunoblotting. The following antibodies were purchased: mouse anti-α-Tubulin (clone 371 DM1A, ab7291, Abcam); mouse anti-Flag (M2, F1804, Sigma-Aldrich); rabbit anti-WNT5A/B (2530, Cell Signaling Technology); rabbit anti-LRP6 (3395, Cell Signaling Technology); rabbit anti-DVL2 (3216, Cell Signaling Technology); IRDye 800CW goat anti-rabbit IgG (926-32211, Li-Cor); IRDye 800CW goat anti-mouse IgG (926-32210, Li-Cor); IRDye 680RD goat anti-mouse IgG (926-68070, Li-Cor); IRDye 800CW goat anti-mouse IgG (926-32210, Li-Cor). For immunoblotting, commercial primary antibodies were used at 1/1000 dilution, except for anti-DVL2, which was used at 1/500. Secondary antibodies were used at 1/30,000. The recombinant anti-FZ antibody F2.A was described previously (*Pavlovic et al., 2018*). The recombinant anti-FZ1 antibody F1.C was generated using phage display methods similar to those described previously (PCT/US2014/051070; *Pavlovic et al., 2018*). Briefly, Fab-phage pools were cycled through four rounds of selection for binding to Fc-fused human FZ1 CRD (R&D Systems) and clonal phages were screened by ELISA for selectivity for FZ1 CRD over the closely related FZ2 and FZ7 CRDs. Variable domains from FZ1-selective clones were cloned into mammalian expression vectors to enable expression and purification of antibody F1.C in the human IgG1 format, as described (*Pavlovic et al., 2018*). Isotype control antibodies were generated against the Gaussia luciferase protein.

### Western blotting

Protein lysates for SDS-PAGE and western blotting were prepared in 1.5× LDS sample buffer (NP0007, Life Technologies). All protein lysates were heated at 95°C for 10 min before SDS-PAGE and western blotting. Quantitative western blotting was performed using the Odyssey DLx (Li-Cor) or Sapphire (Azure BioSystems) infrared scanner according to the manufacturer's instructions. DVL2

phosphorylation was quantified via ImageJ using the raw images generated from the Odyssey DLx scans.

## Recombinant proteins and inhibitors

The following recombinant proteins and drug were purchased: human/mouse WNT5A (654-WN-010, R&D Systems); Wnt-C59 (C7641-2s; Cellagen Technology).

## WNT5A-ROR signaling assays

For flow cytometry-based signaling assay, iMEF cells expressing the GFP-Pdzrn3 reporter were plated at a density of 0.02 million/well in 48-well plates 3 days before WNT5A stimulation. Once adhered to the plate (~1 hr after plating), cells were incubated with Wnt-C59 (100 nM final concentration) and allowed to reach confluency. Wnt-C59 is an inhibitor of the membrane-bound *O*-acyltransferase Porcupine (*Proffitt et al., 2013*) and was used in the experiments to block the activity of endogenous WNTs. 72 hr after plating, cells were stimulated with either WNT5A or an equivalent volume of the control buffer (PBS with 0.1% BSA) in the presence of 100 nM Wnt-C59 for 6 hr. Cells were then harvested with trypsin, neutralized with complete culture media, resuspended in PBS+0.5% FBS, and analyzed using a flow cytometer (Cytoflex, Beckman Coulter) (*Karuna et al., 2018*). Raw data were acquired and analyzed using the CytExpert software (Beckman Coulter). Processing entailed gating out dead cells, calculation of median fluorescence, and correction of the GFP-Pdzrn3 reporter signals against the intrinsic autofluorescence of the cells. For anti-FZ whole IgG and Fab treatments without WNT5A, the antibodies were added for 6 hr before flow analysis. For F2.A Fab inhibition of WNT5A-induced signaling, reporter cells were pretreated with the F2.A Fab for 30 min and then stimulated with WNT5A for 6 additional hours; the F2.A Fab was maintained throughout the WNT5A stimulation period.

For assaying DVL2 phosphorylation by western blotting, 0.08 M cells were seeded per 12-well and cultured in the presence of 100 nM WNT-C59 until completely confluent (3–4 days). A range of WNT5A concentrations (50–200 ng/ml, or 1.3–5.3 nM) was initially tested. 100 ng/ml (2.6 nM) was found to provide the best dynamic range for DVL2 phosphorylation and was used for all subsequent experiments (shown in *Figure 3—figure supplement 2*, *Figure 3—figure supplement 3*, *Figure 4—figure supplement 1*, and *Figure 5—figure supplement 1*).

## Surface staining of Flag-tagged ROR2

Cells were grown to confluency in six-well plates and harvested using Accutase (A1110501, Thermo Scientific). The Accutase was neutralized with complete media and then removed by centrifugation. All subsequent steps were performed on ice. Cells were blocked in PBS+5% goat serum for 20 min, incubated with 0.5 µg/ml anti-Flag antibody diluted in PBS+5% goat serum for 1 hr and then washed 3× with PBS. Cells were then incubated with 1 µg/ml Alexa Fluor 647 goat anti-mouse secondary antibody for 1 hr, washed 3× with PBS, resuspended in PBS+0.5% FBS and analyzed by flow cytometry. ROR KO cells without rescue and ROR KO cells rescued with WT ROR2 were used as negative and positive controls, respectively.

## Endo H and PNGase F sensitivity assay

Protein lysates were collected from cells grown to confluency in six-well plates using 1× radioimmunoprecipitation buffer (89900, Pierce, Thermo Scientific) supplemented with protease inhibitors (78437, Halt protease inhibitor cocktail, 100×, Thermo Scientific). 120 µg of total protein, in a volume of 60 µl was first mixed with glycoprotein denaturation buffer and incubated at 98°C for 10 min. The reaction was then split into three aliquots (36 µg per aliquot in 18 µl) for mock (with no enzyme), Endo H (P0704S, New England Biolabs) and PNGase F (P0702S, New England Biolabs) treatments. For enzyme treatments, the 18 µl of denatured lysate was mixed with 2.5 µl of respective reaction buffer and 2.5 µl of enzyme. For PNGase F treatment, 2 µl of NP-40 was added to each reaction. The reaction volume was brought up to 25 µl for all reactions and incubated at 37°C for 2 hr. The samples were analyzed by western blotting for LRP6, ROR2, and Tubulin.

## WNT5A-CRD binding assay

DNA fragments encoding the ROR2 CRD (residues 170–311), FZ2 CRD (40–165), and FZ7 CRD (45–170) were subcloned into a modified pCX-Fc vector to express CRD fusion proteins that are N-terminally

fused to the signal sequence of human IgG and the Flag tag and C-terminally fused to the Fc fragment of human IgG. The native signal sequence of ROR2, FZ2, and FZ7 were removed during subcloning. As a negative control for WNT5A binding, a construct containing the signal sequence of human IgG, the Flag tag, and the Fc fragment, but no CRD, was generated. All constructs were confirmed by Sanger sequencing. The resulting constructs were transiently transfected into HEK293T cells and cultured in DMEM supplemented with 10% fetal bovine serum, glutamine, and penicillin/streptomycin. 16 hr after transfection, culture media were replaced with DMEM supplemented with 10% KnockOut serum replacement (10828010; Thermo Scientific), glutamine, and penicillin/streptomycin, and conditioned for 24 hr. KnockOut serum replacement was used to avoid the IgG in fetal bovine serum from being carried over into the subsequent Fc fusion-protein A bead binding reactions. After collection, the conditioned media (~9 ml), which were quite acidic based on their yellowish appearance, were neutralized with 20 µl of 1.5 M Tris pH 8.8 and centrifuged to remove cell debris. A pilot binding experiment was performed using Protein A Plus UltraLink beads (53142; Thermo Scientific) to determine the relative expression of the Fc proteins. This information was used to normalize the volume of each conditioned media to be used in the final WNT5A binding experiment such that the molar ratio of each Fc fusion protein on the beads is approximately equal. For the WNT5A pulldown experiment, normalized volumes of conditioned media were incubated with 25 µl packed volume of Protein A-Sepharose beads at 4°C for 16 hr with rotation. The beads were washed 3× with cold PBS and then incubated with 540 ng of purified recombinant WNT5A per Fc construct in 1.5 ml of PBS+5 mg/ml BSA. The binding reactions were incubated at 4°C for 16 hr, after which the beads were washed 3× with cold PBS. Proteins bound to the beads were eluted by heating at 95°C for 10 min in 2× Laemmli buffer supplemented with β-mercaptoethanol and analyzed by western blotting. To minimize potential cross-reactivity of the secondary antibody toward the Fc portion of the fusion proteins on western blots, the goat anti-rabbit IRDye 800CW antibody used in the experiment was pre-absorbed against Fc immobilized on a nitrocellulose membrane. The anti-WNT5A antibody shows some cross-reactivity to ROR2 CRD, which explains the higher molecular weight (~65 kD) band seen in the uncropped blot of *Figure 4A*.

## Acknowledgements

We thank members of the Siebold and Ho labs for their input and discussions. We thank the staff of Diamond Light Source UK beamlines I03, I23, and B21 for assistance (MX14744 and MX19946), and K El Omari, T Walter, K Harlos, and D Staunton for technical support. We thank Michael Greenberg and Linda Hu (Harvard Medical School) for providing anti-ROR2 antibodies. We thank Gen Wen Lim, Claudia Wang, Morgan Krueger, Ember Hung, and Vaishanavi Saware for experimental assistance; Bridget McLaughlin and Jonathan Van Dyke at the UC Davis Cancer Center Flow Cytometry core for training and technical assistance (supported by National Cancer Institute 2P30CA093373-19 and S10 OD018223). This work was supported by the National Institutes of Health (1R35GM119574, 1R35GM144341) to HHH, and Cancer Research UK (C20724/A26752 and DRCRPG-May23/100002) to CS. SCG was supported by a Wellcome Trust-funded DPhil studentship (099675/Z/12/Z). Additional support from the Wellcome Trust Core Award Grant Number 203141/Z/16/Z is acknowledged.

---

## Additional information

### Funding

| Funder | Grant reference number | Author |
| --- | --- | --- |
| National Institutes of Health | 1R35GM119574 | Hsin-Yi Henry Ho |
| National Institutes of Health | 1R35GM144341 | Hsin-Yi Henry Ho |
| Cancer Research UK | C20724/A26752 | Christian Siebold |
| Wellcome Trust | 10.35802/203141 | Christian Siebold |

| Funder | Grant reference number | Author |
|---|---|---|
| Wellcome Trust | 10.35802/099675 | Samuel C Griffiths |
| National Institutes of Health | 2P30CA093373-19 | Hsin-Yi Henry Ho |
| Cancer Research UK | DRCRPG-May23/100002 | Christian Siebold |
| National Institutes of Health | S10 OD018223 | Hsin-Yi Henry Ho |

The funders had no role in study design, data collection and interpretation, or the decision to submit the work for publication. For the purpose of Open Access, the authors have applied a CC BY public copyright license to any Author Accepted Manuscript version arising from this submission.

## Author contributions

Samuel C Griffiths, Conceptualization, Data curation, Formal analysis, Validation, Investigation, Visualization, Methodology, Writing – original draft, Project administration, Writing – review and editing; Jia Tan, Conceptualization, Data curation, Formal analysis, Validation, Investigation, Visualization, Methodology, Writing – original draft, Project administration; Armin Wagner, Data curation, Formal analysis, Supervision, Validation, Investigation, Methodology; Levi L Blazer, Jarrett J Adams, Resources, Validation, Investigation, Methodology; Srisathya Srinivasan, Data curation, Formal analysis, Supervision, Validation, Investigation, Methodology, Writing – review and editing; Shayan Moghisaei, Data curation, Formal analysis, Validation, Investigation; Sachdev S Sidhu, Conceptualization, Resources, Data curation, Formal analysis, Supervision, Funding acquisition, Validation, Investigation, Visualization, Methodology, Writing – original draft, Project administration, Writing – review and editing; Christian Siebold, Hsin-Yi Henry Ho, Conceptualization, Data curation, Formal analysis, Supervision, Funding acquisition, Validation, Investigation, Visualization, Methodology, Writing – original draft, Project administration, Writing – review and editing, Resources

## Author ORCIDs

Samuel C Griffiths http://orcid.org/0000-0002-9257-7354
Jia Tan http://orcid.org/0000-0002-2079-6937
Armin Wagner http://orcid.org/0000-0001-8995-7324
Levi L Blazer http://orcid.org/0000-0001-9594-4642
Srisathya Srinivasan http://orcid.org/0000-0002-4502-5063
Sachdev S Sidhu https://orcid.org/0000-0001-7755-5918
Christian Siebold http://orcid.org/0000-0002-6635-3621
Hsin-Yi Henry Ho https://orcid.org/0000-0002-8780-7864

## Ethics

This study was performed in strict accordance with the recommendations in the Guide for the Care and Use of Laboratory Animals of the National Institutes of Health. All of the animals were handled according to approved institutional animal care and use committee (IACUC) protocols (#21394) of the University of California, Davis. The protocol was approved by the IACUC of the University of California, Davis.

## Decision letter and Author response

Decision letter https://doi.org/10.7554/eLife.71980.sa1
Author response https://doi.org/10.7554/eLife.71980.sa2

# Additional files

## Supplementary files

• Supplementary file 1. Data collection and refinement statistics for ROR2 CRD-Kr. Data in parenthesis refer to highest resolution shell unless otherwise stated. RMSD, root mean square deviation.

• Supplementary file 2. Evolutionary analysis of cysteine-rich domain (CRD) structures. [A]PDB accession codes displayed in parenthesis; [B]RMSD (root mean square deviation) values were

calculated for equivalent Cα atom positions using the program SHP (*Stuart et al., 1979*; *Riffel et al., 2002*); [C]Number of equivalent Cα positions used in calculation of RMSD values with SHP; [D]Summed structural correlation (total probability) values calculated via SHP. The phylogenetic tree for CRDs analyzed (*Figure 1C*) was assembled using PHYLIP (*Felsenstein, 1989*). These summed structural correlation values were used to construct a distance matrix. FZ8-PAM – Frizzled 8-palmitoleate complex (*Janda et al., 2012*), Smo – Smoothened (*Byrne et al., 2016*), FZ8 – Frizzled 8 (*Dann et al., 2001*), sFRP3 – secreted Frizzled-related protein 3 (*Dann et al., 2001*), MuSK – muscle-specific kinase (*Stiegler et al., 2009*), NPC1 – Niemann-Pick C1 protein (*Kwon et al., 2009*), RFBP – riboflavin-binding protein (*Monaco, 1997*), FRα – folate receptor α (*Chen et al., 2013*), FRβ – folate receptor β (*Wibowo et al., 2013*), JUNO – folate receptor δ (*White et al., 2016*). FZ7 – Frizzled 7 (*Nile et al., 2017*).

- Transparent reporting form
- Source data 1. Original and uncropped gel and blot images.

## Data availability

All data generated or analysed during this study are included in the manuscript and supporting files. Atomic coordinates and structure factors for the ROR2 CRD and Kringle domain were uploaded to the PDB under accession code 9FSE.

The following dataset was generated:

| Author(s) | Year | Dataset title | Dataset URL | Database and Identifier |
|---|---|---|---|---|
| Griffiths SC, Tan J, Wagner A, Blazer LL, Adams JJ, Srinivasan S, Moghisaei S, Sidhu SS, Siebold C, Ho HH | 2024 | Human ROR2 cysteine-rich domain (CRD) and Kringle domain | https://www.rcsb.org/structure/9FSE | RCSB Protein Data Bank, 9FSE |

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
