## [Editor Report]

This manuscript describes the crystal structure of the extracellular portion of the ROR2 cell surface receptor, which plays important roles in development and disease. The work provides valuable new insights into the mechanism by which WNTs interact with cell surface receptors to activate downstream signaling events. The insights are clear, and the supporting data are convincing.

---

## [Decision Letter]

**Decision letter after peer review:**

Thank you for submitting your article "Structure and function of the ROR2 cysteine-rich domain in vertebrate noncanonical WNT5A signaling" for consideration by *eLife*. Your article has been reviewed by 3 peer reviewers, one of whom is a member of our Board of Reviewing Editors, and the evaluation has been overseen by Olga Boudker as the Senior Editor. The following individuals involved in review of your submission have agreed to reveal their identity: Karl Willert (Reviewer #2); Neil McDonald (Reviewer #3).

Essential revisions (for the authors):

1. The ROR2 domain deletion analysis in Figure 3 is a critical feature of the paper. The claim that these constructs are expressed at comparable levels is not apparent from the western blot (which is quite dirty). More importantly, the authors need to show cell surface levels (e.g. by flow cytometry) to verify that these mutants are present at comparable surface levels; otherwise interpreting the reporter assay is not feasible. Likewise, the mutants shown in Figure 4A migrate differently than WT, which suggests that the non-signaling mutants are not fully processed.

2. A key conclusion is that WNT5A signaling requires only a membrane tethered CRD from ROR2 and Fzd. However, it is unclear why expression of ∆FZ-CRD and ∆Fz-CRD-Kr would lower basal signaling activities. The authors propose that basal signaling levels in ROR DKO cells are due to WNT5A's ability to signal solely through FZD. The finding that non-functional ROR2 proteins lower this signal suggest dominant negative activity of these proteins on WNT5A-FZD signaling. These observations need to be highlighted and addressed, for example, using targeted deletion or knockdown of the most highly expressed FZD gene in these cells.

3. The finding that the FZD-blocking antibody abrogates all WNT5A signaling activity is impressive. However, given the importance of this result to the final conclusion that WNT5A signals through a ROR/FZD receptor complex, there needs to be further validation. For example, does knockdown or knockout of FZD produce the same effect? Targeted deletion or knockdown of the most highly expressed FZD gene in these cells should produce some measurable effect and at least partially lower signaling levels.

It may also be useful to assess whether the pan-specific FZD antibody blocks WNT5A signaling in wildtype cells, i.e. cells that express endogenous ROR1 and 2? Results in Figure 3F predict the answer to be yes, but this is a rescue experiment where exogenous ROR2 expression levels may exceed endogenous expression levels and produce artifactual levels of signaling activity. To extend this question, what level of reporter activity is observed in iMEF+GFP-Pdzrn3 reporter cells prior to 4OHT treatment? These data could be added as a third line in Figure 3B.

4. The authors propose that the ROR2 CRD might work by binding Wnt5A and/or an interaction with the Fzd5 CRD. These may be very weak but at least a qualitative assay (e.g. pulldown and western blot) using the recombinant CRD-Kr with tagged Wnt5A and/or the Fzd5 CRD should be feasible, even if a rigorous binding assay is not.

[Editors' note: further revisions were suggested prior to acceptance, as described below.]

Thank you for resubmitting your work entitled "Structure and function of the ROR2 cysteine-rich domain in vertebrate noncanonical WNT5A signaling" for further consideration by *eLife*. Your revised article has been evaluated by Olga Boudker (Senior Editor) and a Reviewing Editor.

The manuscript has been improved but there are some remaining issues that need to be addressed, as outlined below:

*Reviewer #4 (Recommendations for the authors):*

I am seeing this manuscript for the first time, now in its revised form and with the first round reviews and the authors' response to those reviews in hand. My first reaction is that the authors should count themselves lucky that the reviewers did a careful job and asked a series of critical and probing questions. My second reaction is that the authors have candidly described their experimental challenges (including mis-interpreting some of the original data) and have done an enormous amount of work to improve the manuscript – for example, redoing a large number of assays with epitope tagged proteins and more rigorously analyzing the Robinow mutant ROR proteins.

There are many interesting and important observations in this manuscript. The crystal structure reveals significant differences between mammalian ROR2 and the *Drosophila* Nrk CRD structures – most obviously, absence of a bound lipid in the ROR2 CRD and the absence of a groove that could accommodate a lipid. This observation and the weakness of the observed Wnt-ROR2 binding imply that ROR2 is not likely to be a high affinity Wnt receptor. The structure/function analysis in ROR1/ROR2 double KO cells looks to be technically sound (following improvements based on reviewer comments). Interestingly a minimized version of ROR2, missing almost all of the intracellular domain, appears to be approximately as active as full-length ROR2. The ability of dimerizing anti-FZD antibodies to activate non-canonical signaling in a ROR-dependent manner and the ability of the corresponding Fab to inhibit non-canonical signaling is fascinating. Together with the observation that ROR's ICD is largely or completely dispensable, this suggests that ROR's extracellular domains influence signaling via FZD. Finally, the Robinow point mutations tested, most of which involve loss or gain of a cysteine, mostly affect folding/stability as is the case for many point mutations in human disease.

This work will likely provide the impetus for much future studies of noncanonical Wnt signaling. I think the manuscript is ready for publication.

*Reviewer #5 (Recommendations for the authors):*

In this extensively revised manuscript, the authors have thoroughly addressed the previous critiques and provided some very interesting new data that challenge the current paradigm on the action of Ror receptor in non-canonical Wnt5a signaling. First part of the manuscript validated the finding from recently published studies on fly Ror ortholog, showing that the cysteine-rich domain (CRD) of mammalian Ror2 similarly cannot accommodate binding to the palmitoleic chain of Wnt protein based on protein crystallography. The more significant findings in this manuscript, however, are that (1) surface presentation of the CRD appears to be necessary and sufficient for the signaling activity of Ror2 whereas all other domains, including the entire cytoplasmic tail, are dispensable; and (2) bivalent antibodies against Fz-CRD can mimic Wnt5a to activate downstream target, seemingly in an Ror-independent fashion. These findings are thought-provoking on how Wnt5a/Ror signaling occurs. If I interpreted correctly, they would imply that (1) the key function of Wnt5a is to induce Fz dimerization or oligomerization, in direct contrast to canonical Wnt ligand that signals by bridging Fz with the co-receptor Lrp5/6; (2) Ror is unlikely to function as a signal transducer itself; (3) the ROR2 mutations associated with brachydactyly type B (BDB1), which act in an autosomal dominant fashion and often truncate the entire cytoplasmic tail like the ΔICD mutant in the current manuscript (in Figure 4D-F; Human Molecular Genetics, Vol. 18, p4013-4021), are likely active or even hyperactive for non-canonical Wnt signaling. Although the manuscript did not provide a very conclusive model on how Ror and its CRD function, it did provide valuable contribution to the field with compelling evidence that challenge the current assumptions on the action of various domains of Ror2.

Suggestions to author:

1) It may be helpful to show the density map, difference map and built model around the lipid binding groove as a supplementary figure. It will strengthen the argument that "no bound lipid, nor an internal space that could potentially accommodate it, was observed in our human ROR2 CRD structure".

2) It will help to finalize the model by adding water molecules to the PDB file. Plenty of positive densities appear in the difference map.

3) Figure 3C: ideally, the concentration of Wnt5a should be expressed as nM instead of ng/ ml so that its signaling activity can be easily compared to those of F2A and F1C in Figure 4. In Figure 3C, the EC50 of WNT5A in wild-type cells (ROR cKO) is ~200ng/ ml, which would be ~2 nM and substantially higher than that of F2A and F1C (0.3 and 7 nM, respectively). If this is true, then the result would imply that the signaling activity of Wnt5a is significantly weaker than that of F2A and F1C.

4) Figure 3F and 4B. The dosage of Wnt5a, F2A and F1C IgG used for these assays should be indicated.

5) In Figure 4A (uncropped blot with labels), there is an extra band (~65 kDa) in the Wnt5a pulldown by ROR2 CRD lane, which is not present in all other conditions. What could be the reason for this?

6) The conclusions of the entire manuscript is based on using only GFP-Pdzrn3 as the readout of Wnt5a signaling activation. Some of the key conclusions, such as the domain requirement for Ror2 function (Figure 3F, I) and Ror2 independent signaling activity of F2A and F1C (Figure 4B), could be strengthened further if a second assay (such as endogenous Dvl2 phosphorylation in the author's previous publication (PNAS vol. 109 (11)) was used to confirm the findings with GFP-Pdzrn3.

---

## [Author Response]

Essential revisions (for the authors):1. The ROR2 domain deletion analysis in Figure 3 is a critical feature of the paper. The claim that these constructs are expressed at comparable levels is not apparent from the western blot (which is quite dirty). More importantly, the authors need to show cell surface levels (e.g. by flow cytometry) to verify that these mutants are present at comparable surface levels; otherwise interpreting the reporter assay is not feasible. Likewise, the mutants shown in Figure 4A migrate differently than WT, which suggests that the non-signaling mutants are not fully processed.

We thank the reviewer for the constructive comments. Regarding the western blots, we have further optimized the experimental conditions, including testing different ROR2 antibody preparations, and acquiring a new departmental Li-Cor infrared scanner to improve the signal-to-noise ratio. The endogenous ROR2 protein is expressed at a low level. Despite our best efforts to reduce the expression levels of the ROR2 WT and mutant rescue constructs to approximate that of the endogenous ROR2, going as far as truncating an intron from the EF1 promoter in the lentiviral vector used to drive the transgene, and titrating down the virus titers (see Materials and methods for details), the transgene expression levels are still somewhat higher than the endogenous ROR2. However, we feel it is most important that the expression levels of the mutant proteins are similar to that of the WT ROR2 rescue construct, since interpretation of the mutant signaling data are all based on direct comparison to the WT ROR2 rescue construct, not to the endogenous ROR2. For clarity, we have modified the relevant sentence in the manuscript to “Immunoblotting confirmed that the mutant proteins were expressed at levels comparable to the WT ROR2 rescue construct” (page 10, lines 224 – 225).

Regarding the need to show cell surface levels of the rescue constructs, we agree with the reviewer that this is critical for interpreting the signaling reporter assay. In the original manuscript, all the rescue cell lines were made using untagged ROR2 constructs. However, none of the currently available anti-ROR2 antibodies are suitable for cell surface staining/flow cytometry of untagged ROR2, making it impossible to perform surface staining of the original cell lines. We therefore re-made all the rescue cell lines in the revised manuscript by introducing rescue constructs that were N-terminally (extracellularly) tagged with the Flag epitope (i.e. all of the WT ROR2, domain truncation mutants, mini-ROR2 and Robinow syndrome mutants shown in Figure 3, 4 and 5 of the revised manuscript). We also repeated all the western blots and signaling assays using these new cell lines (Figure 3B, 3C, 3E, 3F, 3H, 3I, 4B, 4E, 4F, 5A, 5B, 5C of the revised manuscript). Importantly, by conducting anti-Flag cell surface staining of live cells and flow cytometry, we confirmed that all the truncation and mini-ROR2 mutants shown in Figure 3 are expressed on the cell surface (Figure 3 Supplement of the revised manuscript).

To address the question of the Robinow mutants migrating differently than WT in SDS-PAGE, we have performed Endo H and PNGase F sensitivity assays and demonstrated that five of the seven mutants are indeed not fully processed (Figure 5C of the revised manuscript). Anti-Flag cell surface staining/flow analysis further showed that these mutants are unable to traffic to the cell surface (Figure 5 Supplement of the revised manuscript). These new findings are very interesting and indicate that the majority of Robinow mutations in CRD and Kr impair secretion, likely caused by protein misfolding. We have included new discussions on how each of these mutations might affect protein conformation based on our structural data (page 14-15, lines 314-335).

2. A key conclusion is that WNT5A signaling requires only a membrane tethered CRD from ROR2 and Fzd. However, it is unclear why expression of ∆FZ-CRD and ∆Fz-CRD-Kr would lower basal signaling activities. The authors propose that basal signaling levels in ROR DKO cells are due to WNT5A's ability to signal solely through FZD. The finding that non-functional ROR2 proteins lower this signal suggest dominant negative activity of these proteins on WNT5A-FZD signaling. These observations need to be highlighted and addressed, for example, using targeted deletion or knockdown of the most highly expressed FZD gene in these cells.

Upon re-making and analyzing the new cell lines that express Flag-tagged ROR2 mutants, and further optimization of assay conditions to minimize experimental noise (e.g. by switching from an old BD FACScan cytometer to a newer Beckman Cytoflex instrument for all assays shown in the revised manuscript), we no longer observed the ∆CRD and ∆CRD-Kr mutants having lower basal signaling activities in the new experiments (Figure 3F of the revised manuscript).

3. The finding that the FZD-blocking antibody abrogates all WNT5A signaling activity is impressive. However, given the importance of this result to the final conclusion that WNT5A signals through a ROR/FZD receptor complex, there needs to be further validation. For example, does knockdown or knockout of FZD produce the same effect? Targeted deletion or knockdown of the most highly expressed FZD gene in these cells should produce some measurable effect and at least partially lower signaling levels.

During manuscript revision, we identified an inadvertent misinterpretation of the original FZ antibody experiment. In the original experiment, we showed that in the presence of the anti-pan FZ CRD antibody (F2.A), WNT5A could no longer cause degradation the GFP-Pdzrn3 reporter, and we concluded that the antibody inhibited the pathway. However, upon re-examination of the raw flow cytometry data, we found that the antibody itself actually activates the pathway, as it lowers the GFP-Pdzrn3 reporter signal even before addition of WNT5A. This explains why we observed that addition of WNT5A on the F2.A pre-treated cells caused no further degradation of the reporter. This also explains why when we calculated the “percent change” in the reporter signal induced by WNT5A, we found that WNT5A caused zero “percent change” when cells were pre-treated with the antibody, whereas in mock (PBS) pre-treated cells, WNT5A caused a strong “percentage change” in reporter activity. In other words, in F2.A pre-treated cells, WNT5A no longer caused reporter degradation, because the reporter is already degraded by the F2.A antibody, not because the antibody blocks WNT5A-induced signaling, as we originally interpreted. We sincerely apologize for the confusion caused by this oversight.

To further corroborate the finding that the FZ antibody can activate signaling, we tested another recombinant antibody (F1.C) that is mono-specific to the FZ1 CRD. We found that F1.C also potently activates the pathway (Figure 4B of the revised manuscript). We performed detailed dose-response analyses for both antibodies and observed that at post-saturating concentrations, both antibodies exhibited reduced efficiency, suggesting that the antibodies likely activate the pathway by dimerizing FZ (Figure 4C and Figure 4D of the revised manuscript). To further explore this idea and probe the functional requirement of FZ in WNT5A-induced signaling, we tested the monomeric Fab fragment of F2.A, which should compete with WNT5A for FZD CRD binding, but not dimerize FZ. Consistent with this prediction, we found that the F2.A Fab fragment no longer activates signaling on its own (Figure 4E of the revised manuscript) but instead acts as an antagonist to inhibit the ability of WNT5A to activate signaling (Figure 4F of the revised manuscript). We believe these analyses using the anti-FZ antibodies strongly support a direct role of FZ in WNT5A signaling (the same conclusion that we had in the original submission).

Regarding genetic deletion of FZ genes in the reporter cells, we agree that this is a direction of great interest.

However, the experiment is exceedingly challenging for a number of reasons. First, the mouse genome contains 10 FZ genes, and according to publicly available data, at least seven are highly expressed at the developmental stage (E12.5) that our iMEFs were derived from (https://www.tcd.ie/Zoology/research/groups/murphy/WntPathway/fzd.php). We have conducted our own expression analysis by RT-qPCR and found that all six of the FZ isoforms targeted by the F2.A antibody (FZ1, 2, 4, 5, 7 and 8) are expressed in our iMEF reporter cells (Rebuttal Figure 1 below). Specifically, FZ1, 2 and 7 are expressed at levels higher than ROR2, and FZ4 and 8 are expressed at levels similar to ROR1. Therefore, it would be necessary to knock down or knock out most, if not all, of these FZs, which would be a major undertaking. Second, our iMEFs are nonclonal and heterogenous in nature, which complicates genetic analyses. CRISPR/Cas9based gene editing would be the best method to conduct FZ loss-of-function study. However, this would require that our reporter cells were clonal, so we could sequence confirm any candidate FZ KO lines and compare their signaling activity to that of the parent clone. We have another ongoing project in the lab that involves knocking out multiple members of another gene family in iMEFs, and in that project the generation and characterization of the compound KO cells alone took nearly two years. Therefore, we feel that a rigorous genetic analysis of the FZ family deserves the full attention of a follow-up project and is beyond the scope of the current study.

**Author response image 1. sa2fig1:** RT-qPCR analysis to quantify expression levels of the six FZ isoforms that are targeted by the F2.A antibody. WT iMEF RNA was used for the analysis. ROR2 and ROR1 were included for comparison. The expression levels of all tested genes are normalized to that of ROR2. Primer sequences and validation data are shown in Author response table 1.

**Author response table 1. sa2table1:** 

Gene	Primer sequence	R^2^ value	PCR efficiency (%)
ROR2	5’ CGACGCACAGCCCGAACCAC 3’5’ GAAGGATGGGATGGCAAACT 3’	0.9932	114.3
ROR1	5’ CAAATGGCAAGAAAGTGGTG 3’5’ CGGCTGACAGAATCCATCTT 3’	0.9826	95.2
FZ1	5’ CCCATGAGCCCAGACTTTAC 3’5’ CTGTTGGTAAGCCTCGTGTAG 3’	0.9911	100.9
FZ2	5’ CTTCACGGTCACCACCTATTT 3’5’ AACGAAGCCCGCAATGTA 3’	0.9909	115.2
FZ4	5’ GTGGATGCCGATGAACTGA 3’5’ GTCTGTCTTTGTCCCGTCTTT 3’	0.9947	105.6
FZ5	5’ GTCACACCCACTCTACAACAA 3’5’ GAGATGAAGCACAGCACAGA 3’	0.9965	99.2
FZ7	5’ CGGTCAAGACAATCACCATT 3’5’ GATGAAGAGGTAGACGAACAAGG 3’	0.9928	106.8
FZ8	5’ CCGAATCCGTTCAGTCATCAA 3’5’ GGCATGGGCAGTTGTGCGTG 3’	0.986	101.7

It may also be useful to assess whether the pan-specific FZD antibody blocks WNT5A signaling in wildtype cells, i.e. cells that express endogenous ROR1 and 2? Results in Figure 3F predict the answer to be yes, but this is a rescue experiment where exogenous ROR2 expression levels may exceed endogenous expression levels and produce artifactual levels of signaling activity. To extend this question, what level of reporter activity is observed in iMEF+GFP-Pdzrn3 reporter cells prior to 4OHT treatment? These data could be added as a third line in Figure 3B.

Using the Fab fragment of the F2.A pan-specific FZ antibody, we have now shown that it can completely block WNT5A-induced signaling in WT cells (i.e. ROR conditional KO (cKO) iMEF reporter cells prior to 4OHT treatment, which express only endogenous ROR2; Figure 4F of the revised manuscript).

In addition, as suggested by the reviewer, we have repeated the WNT5A dose-response analysis to show the endogenous Wnt5a-Ror activity in iMEF+GFP-Pdzrn3 reporter cells prior to 4OHT treatment (i.e. in the cKO cells; the black curve in Figure 3C of the revised manuscript).

4. The authors propose that the ROR2 CRD might work by binding Wnt5A and/or an interaction with the Fzd5 CRD. These may be very weak but at least a qualitative assay (e.g. pulldown and western blot) using the recombinant CRD-Kr with tagged Wnt5A and/or the Fzd5 CRD should be feasible, even if a rigorous binding assay is not.

We have included in the revised manuscript a new pulldown experiment to directly compare the binding affinities of WNT5A to the CRDs of ROR2, FZ2 and FZ7. We detected robust binding of WNT5A to FZ2 and FZ7 CRDs. Importantly, we also observed that despite lacking the palmitoleate binding site (as seen in our crystal structure), the ROR2 CRD can still bind weakly to WNT5A (Figure 4A of the revised manuscript). This is a very interesting finding. We proposed in the original manuscript that the ROR2 CRD could possibly interact with WNT5A thorough a weak binding site analogous to the “Site 2” observed in the WNT8-FZ8 structure (PDB ID 4F08, PMID: 22653731). Our new WNT5A binding data support this model and further suggest that one molecule of WNT5A could potentially interact with both the ROR2-CRD and FZ-CRD, thereby bringing the two receptors into close proximity (Figure 6 of the revised manuscript).

[Editors’ note: what follows is the authors’ response to the second round of review.]

The manuscript has been improved but there are some remaining issues that need to be addressed, as outlined below:Reviewer #5 (Recommendations for the authors):In this extensively revised manuscript, the authors have thoroughly addressed the previous critiques and provided some very interesting new data that challenge the current paradigm on the action of Ror receptor in non-canonical Wnt5a signaling. First part of the manuscript validated the finding from recently published studies on fly Ror ortholog, showing that the cysteine-rich domain (CRD) of mammalian Ror2 similarly cannot accommodate binding to the palmitoleic chain of Wnt protein based on protein crystallography. The more significant findings in this manuscript, however, are that (1) surface presentation of the CRD appears to be necessary and sufficient for the signaling activity of Ror2 whereas all other domains, including the entire cytoplasmic tail, are dispensable; and (2) bivalent antibodies against Fz-CRD can mimic Wnt5a to activate downstream target, seemingly in an Ror-independent fashion. These findings are thought-provoking on how Wnt5a/Ror signaling occurs. If I interpreted correctly, they would imply that (1) the key function of Wnt5a is to induce Fz dimerization or oligomerization, in direct contrast to canonical Wnt ligand that signals by bridging Fz with the co-receptor Lrp5/6; (2) Ror is unlikely to function as a signal transducer itself; (3) the ROR2 mutations associated with brachydactyly type B (BDB1), which act in an autosomal dominant fashion and often truncate the entire cytoplasmic tail like the ΔICD mutant in the current manuscript (in Figure 4D-F; Human Molecular Genetics, Vol. 18, p4013-4021), are likely active or even hyperactive for non-canonical Wnt signaling. Although the manuscript did not provide a very conclusive model on how Ror and its CRD function, it did provide valuable contribution to the field with compelling evidence that challenge the current assumptions on the action of various domains of Ror2.Suggestions to author:1) It may be helpful to show the density map, difference map and built model around the lipid binding groove as a supplementary figure. It will strengthen the argument that "no bound lipid, nor an internal space that could potentially accommodate it, was observed in our human ROR2 CRD structure".

As suggested by the Reviewer, we have now added three additional figure panels to supplementary Figure 1 (F-H). The final structure refined using PHENIX and corresponding σ-A weighted 2Fo-map (+1σ, blue) and Fo-Fc map (+/- 3.5σ, green/red) calculated at 2.48 Å resolution clearly reveal that the palmitoleate binding pocket on the ROR2 CRD is occluded by a ROR2 loop stretching from M212-T219.

2) It will help to finalize the model by adding water molecules to the PDB file. Plenty of positive densities appear in the difference map.

We have now carefully reprocessed the collected X-ray data using the program autoPROC, and also applying an anisotropy correction using the inbuilt STARANISO module (PMID 21460447). This has resulted in improved data collection statistics and an increase of resolution from 2.7 Å to 2.48 Å, as well as an improved structural model and σ-A weighted 2Fo-Fc and Fo-Fc maps with no prominent positive densities (Author response image 2). This new model fully supports our previous results and observations, and also allowed us to address the Reviewer’s comments about building a water model. We have now included 11 water molecules; some also observed in the CRD-Kr interface. We have also added these new results to the text and included a new panel in Supplementary Figure 1 (panel E).

**Author response image 2. sa2fig2:** Map comparison of the newly processed data at 2. 48 Å resolution from autoPROC+STARANISO (A) compared to the map originating from the previous data processing method, obtained using Xia2+3dii (2.7 Å resolution) (B). View is of the CRD-Kr interface. Two water molecules (Wat-1 and Wat-2) in the interface are clearly visible in the new map, but not in the old map. In both panels the final refined model is shown. The maps shown are σ-A weighted 2Fo-maps, contoured at 1σ.

3) Figure 3C: ideally, the concentration of Wnt5a should be expressed as nM instead of ng/ ml so that its signaling activity can be easily compared to those of F2A and F1C in Figure 4. In Figure 3C, the EC50 of WNT5A in wild-type cells (ROR cKO) is ~200ng/ ml, which would be ~2 nM and substantially higher than that of F2A and F1C (0.3 and 7 nM, respectively). If this is true, then the result would imply that the signaling activity of Wnt5a is significantly weaker than that of F2A and F1C.

In the revised Figure 3C, we have included both nM and ng/ml as the WNT5A concentration units in the X-axis. It is correct that the EC_50_ of WNT5A appears to be much lower than that of F2.A and F1.C. We believe this is because WNT proteins are notoriously difficult to produce recombinantly, and therefore the specific activity of the WNT5A that we purchased from R&D is likely lower than that of the native protein. This makes it difficult to directly compare the potency of WNT5A with that of F2.A and F1.C based on protein concentrations.

4) Figure 3F and 4B. The dosage of Wnt5a, F2A and F1C IgG used for these assays should be indicated.

We have revised Figure 3F and Figure 4B legends to more clearly indicate the concentration of these proteins used in the assay. (Lines 996; 1038-1039).

5) In Figure 4A (uncropped blot with labels), there is an extra band (~65 kDa) in the Wnt5a pulldown by ROR2 CRD lane, which is not present in all other conditions. What could be the reason for this?

Based on our experience from several rounds of prior optimization experiments, this extra band is most certainly caused a cross-reactivity of the anti-WNT5A antibody (Catalog #2530, Cell Signaling Technology) toward the ROR2 CRD. The ~65 kD band is exactly where the ROR2 CRD-Fc runs, and the same anti-WNT5A antibody also cross-reacts with a ROR2 CRD-Kr-Fc protein that we blotted in a separate experiment. We have now included this explanation in the Methods section (Lines 668-670), as well as with the uncropped blot image.

6) The conclusions of the entire manuscript is based on using only GFP-Pdzrn3 as the readout of Wnt5a signaling activation. Some of the key conclusions, such as the domain requirement for Ror2 function (Figure 3F, I) and Ror2 independent signaling activity of F2A and F1C (Figure 4B), could be strengthened further if a second assay (such as endogenous Dvl2 phosphorylation in the author's previous publication (PNAS vol. 109 (11)) was used to confirm the findings with GFP-Pdzrn3.

We have performed new western analyses to show that the key conclusions concerning the domain requirement for ROR2 function (Figure 3F, I) and ROR2 independent signaling activity of F2.A and F1.C (Figure 4B) hold true when we assayed endogenous DVL2 phosphorylation as a second readout of WNT5A-ROR signaling. These new data are included in Figure 3—figure supplement 2, Figure 4—figure supplement, and Figure 5—figure supplement 1 of the revised manuscript.